# From Grunts to Lexicons: Emergent Language from Cooperative Foraging

## Abstract

Language is a fundamental tool for human cooperation, yet its evolutionary origins remain an open question. One leading hypothesis suggests that language emerged to meet the social and ecological demands of early human survival. Inspired by this view, we investigate the emergence of communication in Foraging Games (FG), a multi-agent framework designed to reflect the cognitive and environmental constraints of cooperative foraging. In FG, agents operate in a shared, partially observable grid world and must coordinate through bidirectional communication to solve tasks involving hidden rewards and temporal dependencies. Using end-to-end deep reinforcement learning, agents learn physical actions and communication strategies from scratch without shared parameters. Our results show that agents develop protocols exhibiting hallmark features of natural language: arbitrariness, interchangeability, displacement, cultural transmission, and compositionality. We quantify these properties and analyze how population size, social dynamics, and temporal constraints shape emergent linguistic structures. Our framework provides a platform for studying the evolution of language in embodied, multi-agent settings.

## 1. Introduction

The evolution of human language remains a central open question in science. While humans fluently use and understand language, its origins are still debated. One influential view proposes that language emerged through cooperative interaction under partial observability (Sterelny, 2012; Tomasello, 2010; Nowak et al., 2000; Christiansen & Kirby, 2003; Nowak & Komarova, 2001). Rather than functioning as an abstract code, language is understood as

a tool shaped by social use and shared goals (Wittgenstein, 2009; Wagner et al., 2003). Although direct evidence from early stages is unavailable, multi-agent simulations offer a way to examine how ecological and social conditions might have shaped language (Cangelosi & Parisi, 2012; Kirby et al., 2014; Lazaridou et al., 2017; 2018).

Prior work on Emergent Communication (EC) has primarily focused on referential games (RG), where a speaker conveys task-relevant information to a listener over a limited communication channel (Lazaridou et al., 2017; Lewis, 2008; Kharitonov et al., 2019; Gualdoni et al., 2024). These settings have advanced our understanding of symbol grounding and compositionality, but often impose simplifying assumptions: agents are limited to unidirectional communication (Galke et al., 2022), operate in fixed speaker-listener roles, and perform disembodied tasks with passive input processing and no active interaction with the environment. Even recent studies on population-based learning (Dubova & Moskvichev, 2020; Dubova et al., 2020; Kim & Oh, 2021; Chaabouni et al., 2022; Michel et al., 2023) decouple communication from physical behavior. Such setups diverge from the embodied conditions under which human language likely evolved (Tomasello, 2010; Dessalles, 2007). The pioneering work integrates embodiment into EC (Mordatch & Abbeel, 2018) and reveals richer aspects of emergent language, including the co-learning of physical and linguistic behavior and the use of bodily actions as implicit communication. However, it does not incorporate biologically plausible decentralized learning, agent heterogeneity, or population structure, limiting its ability to support broader studies of EC in embodied settings.

Following this line of work (Mordatch & Abbeel, 2018), we introduce Foraging Games (FG), a multi-agent framework designed to reflect more realistic ecological and cognitive constraints that may have shaped early human language (Dessalles, 2007; Tomasello, 2010). FG supports bidirectional communication. Agents are trained to jointly learn both physical action and communication, as early humans likely did during cooperative foraging. The environment enforces embodiment: agents must explore, observe, and act within a dynamic and partially observable world that they can influence. Moreover, each agent interacts with a population of diverse partners,

[1]Anonymous Institution, Anonymous City, Anonymous Region, Anonymous Country. Correspondence to: Anonymous Author <anon.email@domain.com>.

Preliminary work. Under review by the International Conference on Machine Learning (ICML). Do not distribute.

**Figure 1. An Overview of Foraging Games.** (a) Two agents operate in a $5 \times 5$ partially observable grid world. Each agent can observe a $3 \times 3$ grid centered on itself. The two agents are required to pick up the goal items simultaneously. (b) *ScoreG* game: Two items are assigned random scores; each agent observes only one score, and both must pick up the higher-scoring item to succeed in an episode. This game is designed to encourage agents to communicate about items' scores. (c) *TemporalG* game: Two items spawn at random times, each observed by only one agent. Agents must collect them in chronological order, encouraging communication about time.

facilitating the study of generalization to new partners, dialect formation, and cultural transmission (Kim & Oh, 2021; Rita et al., 2022; Michel et al., 2023).

Specifically, we formulate FG as a partially observable grid world in which agents must complete multi-step games as shown in Figure 1. Success depends on both taking effective actions and communicating what they see and know. It includes two games: (1) collecting the most valuable item, which encourages communication about items' scores and locations, and (2) picking up items in a specific order, which requires communication about when items were seen. Each agent has deep neural network modules for perception, memory, and policies to translate internal representations into actions and messages. Agents communicate through discrete messages drawn from a finite set of learnable vocabulary exchanged at each time step. They are trained using Proximal Policy Optimization (PPO) (Schulman et al., 2017), with no shared parameters, or gradients, reflecting the personalized and decentralized nature among agents.

With this framework in place, we aim to provide answers to the following questions: **(Q1)** What properties does emergent communication have when it arises in embodied, interactive, and collaborative agents? **(Q2)** Do agents truly understand the language they use, and what drives the emergence of a shared language when the same agent acts as both speaker and listener? **(Q3)** How do population size and social dynamics affect the emergent language? **(Q4)** Can agents refer to spatial and temporal events through their language? **(Q5)** Can agents communicate implicitly through non-verbal behaviors, such as body movements?

Empirically, we find that agents achieve a success rate (SR) above 95% across all games. Although two trained agents perform well when paired with each other, each fails when paired with a copy of itself at test time, suggesting that an agent understands only its partner's language, not the one it produces. We propose two solutions to this problem. The first is to train a population with more than two agents. We hypothesize that developing a shared language becomes the optimal strategy when agents must communicate with multiple partners. The second is to incorporate self-interaction during training (Dubova & Moskvichev, 2020; Lowe et al., 2020; Charakorn et al., 2023; 2024; Liu et al., 2025), motivated by the observation that humans can speak to themselves. These solutions encourage agents to comprehend their own messages, promoting convergence on a shared common language and reflecting the property of interchangeability. Since human language develops in populations, we further explore language properties in groups of agents connected via different social structures: fully-connected, ring-structured, and small-world-structured networks. We decode task-relevant information, such as items' positions, scores, and spawn times, from the agents' messages using logistic regression, verifying that the messages are meaningful rather than random. Above-chance decoding accuracy indicates that a language has emerged to communicate item properties. Specifically, agents develop time adverbials (Lipinski et al., 2023) that refer to when past events occurred, along with messages indicating the location of those events. Finally, we show that agents can develop implicit communication (Wang et al., 2025; Grupen et al., 2022; Dreyer et al., 2025), i.e., gaining information by observing a partner's behavior.

Our novel contributions are as follows: **(C1)** We introduce Foraging Games, a framework for studying emergent communication that provides ecological and cognitive constraints resembling those faced by early humans, including embodiment, behavior coordination, partial observability, bidirectional communication, and temporal reasoning. **(C2)** We propose a hybrid cross-and-self-play training regime that enhances cultural transmission in weakly-connected social networks: agents that are closer in the training population develop more similar languages than those farther apart. **(C3)** We design a series of tasks that elicit both temporal and spatial displacement in emergent language, which could not arise without embodiment.

## 2. Related Work

EC offers insights for exploring the processes and pressures that might have shaped the evolution of language (Lazaridou et al., 2017; 2018; Kirby et al., 2014; Chaabouni et al., 2022; Cangelosi & Parisi, 1998; Cangelosi, 2001; Boldt

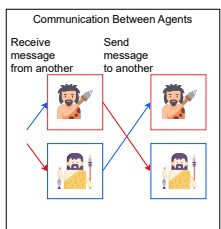

**(a)** Msg Exchange

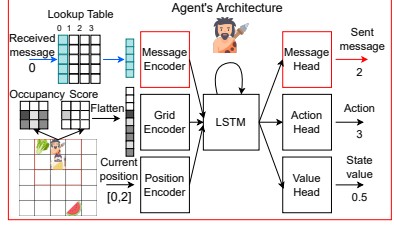

**(b)** Architecture Overview

**Figure 2. A graphical overview of our method.** (a) Agents exchange messages at every time step. (b) The neural architecture of a single agent. On the input side, the received integer message is mapped to a real-valued vector via a learnable lookup table and passed through the message encoder. The grid observation and agent position are processed by the grid encoder and position encoder, respectively. The outputs of all three encoders are concatenated and passed to an LSTM module, which maintains temporal memory. On the output side, the message head, action head, and value head produce the next message token, environment action, and estimated state value.

& Mortensen, 2024; Peters et al., 2025; Lazaridou & Baroni, 2020) and the development of more effective representations (Mu & Goodman, 2021; Carmeli et al., 2025; Zion et al., 2024; Denamganai et al., 2024; Yao et al., 2022). Early studies employed evolutionary computation (Cangelosi & Parisi, 1998; Cangelosi, 2001) or Bayesian modeling (Nowak et al., 2000; Kirby et al., 2007) to simulate language emergence and its dynamics. The advent of deep reinforcement learning (DRL) (Mnih et al., 2013; 2015) introduced a more powerful framework for studying EC in realistic, complex settings (Foerster et al., 2016).

Contemporary work often applies DRL to Lewis-style referential games (RG) (Lewis, 2008), where a speaker conveys task-relevant information to a listener through a limited communication channel, typically for target identification or input reconstruction (Zion et al., 2024; Gualdoni et al., 2024; Lipinski et al., 2024; Kharitonov et al., 2019). Because language is inherently social, research has shifted toward population-level studies (Dubova & Moskvichev, 2020; Dubova et al., 2020; Kim & Oh, 2021; Chaabouni et al., 2022; Rita et al., 2022; Michel et al., 2023), examining how social dynamics and population training influence consistency (Dubova & Moskvichev, 2020; Dubova et al., 2020) and dialect formation (Kim & Oh, 2021). However, many resulting frameworks resemble autoencoders (Kingma et al., 2014; Higgins et al., 2017; Ueda & Taniguchi, 2024), treating language as a latent code rather than a tool for real-world coordination.

Recent studies have examined more realistic conditions, such as bidirectional exchange, where agents both send and receive messages (Evtimova et al., 2018; Dubova & Moskvichev, 2020; Dubova et al., 2020; Taillandier et al., 2023; Graesser et al., 2019; Kottur et al., 2017; Nikolaus, 2024), and embodiment (Jain et al., 2019; Patel et al., 2021; Mordatch & Abbeel, 2018; Kajic et al., 2020; Bullard et al., 2020), where agents learn to communicate through interaction with their environment and partners. Only two studies (Mordatch & Abbeel, 2018; Jain et al., 2019) incorporate both. Most embodied EC work (Jain

et al., 2019; Patel et al., 2021) prioritizes task performance over analyzing EC itself and omits biologically plausible decentralized training. Another line of research uses EC as a control prompt, outperforming natural language in embodied tasks (Mu et al., 2023), though it is still learned through disembodied RG.

Our work builds on the seminal study by Mordatch & Abbeel (2018), which first demonstrated grounded communication in embodied agents. Whereas they used a shared neural network and differentiable environment, we investigate decentralized agents in non-differentiable settings to better mirror human-like learning. Furthermore, we expand the scope to population-level social dynamics and provide quantitative metrics for a broader suite of linguistic properties, including compositionality, interchangeability, cultural transmission, and displacement.

## 3. Foraging Games (FG)

Foraging-style games are used as benchmarks for multi-agent reinforcement learning (MARL) algorithms (Albrecht & Ramamoorthy, 2013; Papoudakis et al., 2021; Yang et al., 2022; Jafferjee et al., 2022; Jaques et al., 2019; Ikram et al., 2021). However, existing environments often assume full observability and are not designed to support the study of emergent communication. We develop our FG, in which decentralized agents share a common goal and must navigate to and simultaneously pick up the goal items to earn equal rewards. Our FG is designed to encourage agents to communicate bidirectionally about their observations. Each agent is provided with only partial knowledge of the environment (e.g., observing only one item). Agents must reach a consensus based on their perspectives by communicating the message from a fixed-size dictionary with learnable embeddings and choosing appropriate actions, where the action space is {move left, move right, move up, move down, and pick up}. FG has two games: *ScoreG* and *TemporalG*. The former is used to investigate four properties of emergent language: **interchangeability** (agents can understand the messages they produce), **arbitrariness** (symbols acquire meaning through social agreement rather

than inherent structure), **compositionality** (messages are built from reusable parts reflecting task semantics), and **cultural transmission** (language patterns are passed across agents), under varying population sizes and training regimes. The latter focuses on **displacement**, aiming to determine whether agents' messages encode temporal and spatial information about seen items in the past.

**Game 1: *Pickup High Score (ScoreG)*** is set in a $2D$ grid world with the size of $5 \times 5$, containing 2 items, 2 agents, and a wall, as shown in Figure 1. The wall refers to the area outside the $5 \times 5$ grid. Each grid cell can be occupied by either an agent or an item, but not both simultaneously. Agents cannot move to the wall, the occupied grid cell, or the same grid cell. Each agent receives an observation $\mathbf{o}^{(t)} = (\mathbf{x}^{(t)}, \mathbf{p}^{(t)})$, where $\mathbf{x}^{(t)} \in \mathbb{R}^{3 \times 3 \times 2}$ (3 is the receptive field size and 2 refers to two channels). The first channel of the receptive field $\mathbf{x}^{(t)}$ represents an occupancy map: a value of zero indicates that a grid cell contains neither an item nor a wall. The second channel shows the item's score. The receptive field of each agent has its center at the agent and observes the agent's surroundings, delineated as a red box in Figure 1. To simulate incomplete knowledge, each agent observes the score of one item during an episode. The agent's absolute position in the environment is given by $\mathbf{p}^{(t)} \in \mathbb{R}^2$. The goal item is defined as the item with the highest score. An episode is deemed successful if both agents navigate to and simultaneously pick up the goal item. In this case, they receive a shared positive reward of $+1$; otherwise, they receive a shared negative reward of $-1$. Additionally, similar to (Wijmans et al., 2020), successful episodes grant both agents a bonus reward $r_b$ based on their efficiency: $r_b = \frac{T_{\max} - T_{\text{total}}}{T_{\max}}$, where $T_{\max} = 10$ is the maximum number of steps allowed per episode, and $T_{\text{total}}$ is the total number of steps taken by the agents. The bonus is to encourage the agents to complete the game as soon as possible. To prevent agents from overfitting to specific scores in the *ScoreG* task while still evaluating in-distribution generalization, we use a held-out set of test scores that do not overlap with those seen during training. Specifically, training scores are randomly sampled from the set $\{5, 10, 15, \ldots, 250\}$, while test scores are sampled from the disjoint set $\{2, 4, 6, 8, 12, \ldots, 248\}$. This setup ensures that test scores lie within the same distributional range as the training scores, while remaining unseen during training. We also include the game variant to unseen item positions in subsection C.5. To isolate the role of explicit communication, agents are made invisible to one another by default, i.e., a grid cell occupied by another agent appears empty. This design prevents any form of implicit communication through visual cues (Wang et al., 2025; Grupen et al., 2022; Li et al., 2021; 2024).

**Game 2: *Pickup Temporal Order (TemporalG)*** Two agents must cooperatively pick up two items by first navigating to and simultaneously collecting the first spawned item, followed by the second. The pickup must occur in the same order as the items appear. The environment follows the same configuration as *ScoreG*, except the agent's observation $\mathbf{x}^{(t)}$ excludes the score channel, and the episode length is limited to $T_{\max} = 20$. Initially, agents are spawned on opposite sides of the grid and remain frozen at time step $t = 1$ to $t = 6$. Subsequently, each item appears at a distinct time step, drawn uniformly from the fixed duration set $\{1, 2, 3, \ldots, 6\}$. Each item is guaranteed to appear within at least one agent's receptive field, ensuring it is visible when it spawns. Agents begin moving and attempting to pick up the items only after $t = 6$. Communication is restricted to adjacent grid cells; otherwise, a zero-value message is transmitted. This constraint induces displacement, compelling agents to retain an event's time and location and later articulate this information once they rendezvous with their partners. Rewards follow the same scheme as in *ScoreG*.

## 4. Experimental Setting

**Embodied Social Agents.** As shown in Figure 2b, each agent in our environment learns both to produce actions and to send and receive discrete messages through a policy trained with PPO (Schulman et al., 2017) (subsection A.1). For an experiment on unidirectional communication, see subsection C.2. We denote the overall policy as $\psi = (\pi, \phi)$, where $\pi$ is the action policy that selects an action, and $\phi$ is the communication policy that selects a discrete message $\mathbf{m}^{(t)}$ to send. We formulate communication as a sequence of discrete messages. At each time step $t$, an agent is only allowed to send one discrete message, where each message is represented by a single integer $\mathbf{m}^{(t)} \in \{0, 1, 2, 3\}$. This integer indexes into a lookup table of 4 learnable embeddings (each of dimension 16) in the receiving agent. Each agent maintains its own message lookup table, which is not shared. In other words, upon receiving a message $\mathbf{m}^{(t)}$, the agent retrieves the corresponding message embedding from its own lookup table. We also report results for vocabulary sizes other than the default of 4 in Table S5.

All agents share the same neural network architecture but are initialized with independent random parameters. At each time step, the agent receives three inputs: a partial grid observation $\mathbf{x}^{(t)}$, its own position $\mathbf{p}^{(t)}$, and the message $\mathbf{m}^{(t-1)}$ from its partner sent in the previous time step (see Figure 2a). Each input is encoded separately, using a multi-layer perceptron for the grid, a linear layer for the position, and a multilayer perceptron for the message. The encoded features are concatenated and passed into a long short-term memory network (LSTM) (Hochreiter & Schmidhuber, 1997), which maintains temporal information in a working memory and outputs a hidden state. This hidden state is used to produce three outputs: a distribution over actions, a distribution over messages, and a scalar value estimate used for computing the PPO advantage

(Schulman et al., 2016). Both the action and the message are sampled from their respective probability distributions predicted by the agent. The message is used to index into the receiver agent's lookup table, retrieving the corresponding embedding, which is then used by the communication policy $\phi$ of the receiver agent to produce the message output for the current time step $t$.

**Training Regimes.** We train each agent independently using standard single-agent PPO, with no shared parameters or centralized critic. Each agent treats its partner as part of the environment and learns solely from its own experience. Communication policies emerge through interaction and reward, without explicit supervision. This decentralized setup is intentional: it allows agents to specialize, diverge, and adapt to others, enabling us to study how communication protocols evolve under varying population sizes, agent heterogeneity, and connectivity patterns among agents. To investigate how communication strategies emerge and generalize in populations, we explore two training regimes that differ in how agents interact with other partners. These regimes allow us to study the effects of population diversity, message alignment, and exposure to self-produced messages on communication development. **Cross-Play Training (XP):** We define the agent population size as $N_{\text{pop}}$. In each training episode, a pair of agents is randomly sampled from the population to play together. **Cross-Play-and-Self-Play Training (XP+SP):** Motivated by this aspect of inner monologue, humans being able to speak to themselves, in each episode, we randomly select either two distinct agents (Cross-Play, XP) or the same agent twice (Self-Play, SP). Specifically, for each episode, we randomly sample agents $i$ and $j$ such that either $i \neq j$ or $i = j$, where $i, j \in \{1, \ldots, N_{\text{pop}}\}$.

**Social Network Structure.** We consider several social network structures (Figure S3): Fully Connected (FC), where every agent interacts with all others; Ring, where each agent interacts only with immediate neighbors; Ring with Cliques (Clq), which adds short-range clusters to the ring; Watts–Strogatz (WS), which rewires a fraction of ring edges to create small-world shortcuts; and Ring with Long-Range Connections (LRC), which augments the ring with distant edges. FC represents the dense baseline. Ring is the sparsest structure while still connected, allowing us to test how communication propagates along long paths. Adding cliques or links (Clq, WS, LRC) reduces social distance and may strengthen language transmission (Appendix B).

**Evaluation Metrics on Language Properties.** To evaluate linguistic properties in emergent communication, we use three standard metrics and a new metric below. Formal definitions of these metrics are provided in subsection A.3. Briefly, **Topographic Similarity (*topsim*)** (Brighton & Kirby, 2006; Lazaridou et al., 2018; Kharitonov et al.,

2019) measures the structural alignment between message space and semantic space. It is computed as the Spearman correlation between pairwise distances in the message space and the corresponding distances in the semantic space, which reflects the agent's observations and context. **Representational Compositionality (*repcom*)** (Elmoznino et al., 2025) quantifies whether semantic representations can be reconstructed from messages using a low-complexity function. Languages achieve higher *repcom* when the mapping function is low-complexity and when it reconstructs the representations accurately, reflecting both simplicity and predictive power. **Language Similarity (*LS*)** (Kim & Oh, 2021; Rita et al., 2022; Michel et al., 2023) measures how similarly two agents communicate in the same situation. It compares the sequences of discrete messages each agent produces and computes the average agreement between them. **Interchangeability (*IC*)** measures whether an agent can understand the language it produces. We define **Self-SR** as the success rate when an agent plays with a copy of itself, and **Cross-SR** as the success rate when paired with a different agent. The overall success rate is denoted as **SR**. *IC* is the ratio of Self-SR to Cross-SR. A high *IC* indicates that agents can better generalize and interpret their own language.

**Message Decoding.** To examine emergent communication properties, we collect each agent's messages over all time steps in every episode. For the *ScoreG* task, messages are concatenated into a single message chain per agent per episode. In the *TemporalG* task, agents are only allowed to communicate when they occupy adjacent grid cells. Thus, we begin collecting messages only after their first adjacency and concatenate all subsequent messages. To decode environmental attributes, such as item scores, spawn times, and positions, from message chains, we use one-vs-rest logistic regression (LR). For example, to decode four possible item locations, we train four binary classifiers, each distinguishing one location from the rest. During inference, the class with the highest predicted probability is selected. Each LR model is trained on 3,500 message chains and tested on 1,500, using 3-fold cross-validation across 3 random seeds. We report message chain decoding accuracy as the average across all agents. Other metrics, including *LS*, *IC*, *topsim*, and *SR*, are computed using 1,000 test episodes.

## 5. Result

We investigate the emergence of linguistic properties in the FG framework according to previously stated research questions (**Q1-Q5**) and present our findings in this section. Each experimental setting is defined by four factors: the game type, agent population size, social structure, and training regime. These factors influence the emergence of different language properties, making exhaustive analysis infeasible. Instead, we report results

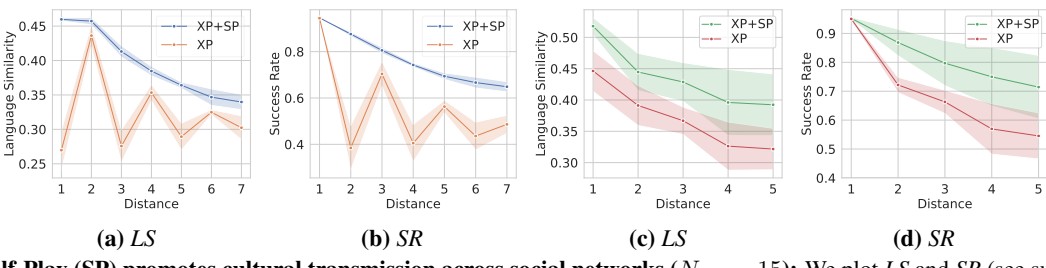

**(a)** *LS*    **(b)** *SR*    **(c)** *LS*    **(d)** *SR*

**Figure 3. Self-Play (SP) promotes cultural transmission across social networks** ($N_{\text{pop}} = 15$)**:** We plot *LS* and *SR* (see subsection A.3) as a function of the shortest-path distance between two agents. (a,b) ring-structured network (Ring). (c,d) ring-structured with clique network (Clq).

from a representative subset of settings. For clarity, we adopt the naming convention for each experimental setting: `[game]-[population size]-[social structure]-[training regime]`. For example, *ScoreG*-P15-Ring-XP+SP refers to a setting where $N_{\text{pop}} = 15$ agents in a ring-structured network are trained with XP+SP regime on the *ScoreG* game.

**Table 1. Cross-play training can cause non-interchangeable languages.** Game performance and language similarity (LS) comparison across different training regimes. $N_{\text{pop}}$ is the population size. *Cross-SR* and *Self-SR* are the mean ($\pm$ standard deviation) success rates when agents play with others or with copies of themselves, respectively.

| Training | $N_{\text{pop}}$ | LS | Cross-SR | Self-SR |
|---|---|---|---|---|
| XP | 2 | $0.215 \pm 0.002$ | $0.987 \pm 0.002$ | $0.065 \pm 0.054$ |
| XP+SP | 2 | $0.598 \pm 0.010$ | $0.968 \pm 0.005$ | $0.968 \pm 0.002$ |
| XP | 3 | $0.527 \pm 0.036$ | $0.977 \pm 0.007$ | $0.944 \pm 0.018$ |

**Increasing population size and self-play training support interchangeable language [*ScoreG*-FC].** Since decentralized agents do not share neural network parameters, it is natural to wonder whether they develop a common language at all. A curious possibility is that one agent might produce L1 while only understanding L2, with the other doing the opposite. Moreover, it is possible that an agent trained with its partner does not understand the language it produces. As shown in Table 1, XP agents trained exclusively with each other fail to understand their own messages when paired with a copy of themselves during test time as indicated by a drop of 6% in *Self-SR*. Surprisingly, when agents are trained with $N_{\text{pop}} = 3$, they maintain high *Self-SR* even without explicit self-play, suggesting that population training encourages the emergence of interchangeable language. These agents also develop more consistent language, as indicated by a higher *LS* of 0.53 compared to the XP alone. Furthermore, XP+SP agents achieve high *Self-SR* due to their direct exposure to SP. Interestingly, they also exhibit greater *LS* across agents, with an *LS* of 0.59, outperforming XP agents even in population settings. Agents with high *LS* produce similar messages, while agents with low *LS* produce distinct ones in the embedding space (Figure S4). In agreement with prior research, (Dubova & Moskvichev, 2020) found that both population training with $N_{\text{pop}} \geq 3$ and the use of SP

facilitate the development of common language.

**Self-play enhances cultural transmission in weakly-connected networks [*ScoreG*-P15].** Human language evolves in sparsely connected societies, giving rise to a shared language alongside dialects and foreign languages. To investigate how language propagates in such settings, we train agents within weakly connected Ring and Clq social networks. The population size is fixed at $N_{\text{pop}} = 15$, with each agent interacting only with a few neighbors. This setup allows us to examine how culture is transmitted across indirect connections at test time. As shown in Figure 3, both *LS* and *SR* generally decline with increasing distance between agents. This indicates that language partially propagates through non-co-trained agents: nearby neighbors develop more similar languages than distant ones. Importantly, the learned language also generalizes to indirectly connected agents, albeit with reduced *SR*. An exception arises for XP agents under the Ring network, which display a distinctive zig-zag pattern in Figure 3a and Figure 3b. Here, an agent produces messages more similar to those of a non-neighbor than to its direct partner (Figure S5c). While agents successfully learn to interpret their neighbors, they fail to produce messages aligned with the language they understand (Figure S5c and Figure S6c), highlighting a breakdown in interchangeability. Furthermore, as shown in Figure 3a and Figure 3c, self-play training yields consistently higher *LS* across all social distances compared to cross-play training. Together, these findings suggest that self-play not only enhances within-agent consistency but also supports more stable and transmissible communication protocols in structured populations. Finally, we find that adding more connections to Ring enhances cultural transmission (Figure S7).

**Spatial and temporal displacement emerge in agent communication [P3-FC-XP].** Humans can refer to past or future events, for example, I saw an apple yesterday or I will collect an apple tomorrow. They can also refer to spatially distant objects, such as an apple in the forest. This ability to refer to things beyond the immediate here and now is known as displacement in natural language. We investigate whether displacement can emerge in artificial foraging agents using the *TemporalG* game. XP agents with

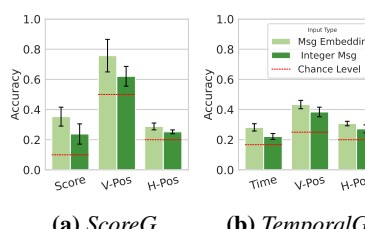

**(a)** *ScoreG*  **(b)** *TemporalG*

**Figure 4. Decoding item states from messages.** *V-Pos*/*H-Pos* are item positions, *Score* is item value, and *Time* is item spawn time. *Integer Msg* uses raw message chains composed of integer sequences. *Msg Embedding* uses chained embeddings mapped from a lookup table. Error bars show standard deviations.

$N_{\text{pop}} = 3$ achieve strong performance in this setting, with a Cross-SR of $0.975$ and a Self-SR of $0.962$ (Table S11). Since the communication range is limited, the agents adopt a rendezvous strategy, meeting near the center of the grid map to establish communication (Figure S11). Next, we decode spatial and temporal displacement information from the messages. The items can take on one of 6 possible spawn times, 4 vertical positions (excluding the grid's center), and 5 possible horizontal positions. The decoding accuracy for the spawn time and position of items is above chance (Figure 4b), suggesting that messages function as temporal adverbials and spatial references. We also decode items' scores and positions from XP agents ($N_{\text{pop}} = 3$) in the *ScoreG* game (Figure 4a). The items can take on one of 10 possible score ranges, 2 possible vertical positions (top and bottom), and 5 possible horizontal positions. The results align with those observed in the *TemporalG* game. Moreover, compared to *Integer Msg*, decoding with chains of message embeddings yields higher accuracy, suggesting that these embeddings encode more meaningful and linearly separable spatial and temporal features. Interestingly, when agents collect targets remotely without movement, their messages stop conveying spatial information (Figure S8), highlighting the role of embodiment in shaping language.

**Implicit communication can emerge when explicit message communication is disabled [*ScoreG*-P3-FC-XP].** We study whether agents can convey information through their sequence of actions in an episode when the explicit communication channel is disabled. We conduct an ablation on two variables: partner visibility and the presence of explicit verbal communication. When a partner is invisible, their location appears as an unoccupied cell in the observing agent's receptive field. We train XP agents in *Inv-NoCom* and *Vis-NoCom*, where agents either have or lack partner visibility, but no explicit message communication is allowed. If an agent observes an extreme score (e.g., 222 or 22), it can confidently infer whether the item is a goal or not. To isolate this confounding factor, we evaluate agents only under the condition that both partners observe high scores. As shown in Figure 5, *Inv-Com* agents (XP agents trained in our default FG environments) achieve SR of $85\%$. Without

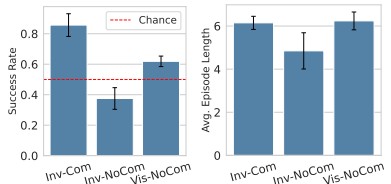

**(a)** Success Rate  **(b)** Avg Ep Length

**Figure 5. Implicit communication in the *ScoreG* games.** *Inv*/*Vis* denote partner visibility; *Com*/*NoCom* indicate presence or absence of verbal communication. Item scores are from $[160, 162, \ldots, 240]$. We report average successful episode length; error bars show standard deviations across seeds and agents.

communication, *Inv-NoCom* agents perform below chance ($40\%$), likely because they fail to coordinate goal pickup without seeing each other. See more failure analysis in subsection C.6. Interestingly, *Vis-NoCom* agents achieve an SR of $60\%$, outperforming *Inv-NoCom*. This indicates that both *Inv-Com* and *Vis-NoCom* agents can glean information from their partners—either via explicit messages or by observing partners' actions. We also evaluate the average length of successful test episodes. Both *Inv-Com* and *Vis-NoCom* agents take around 6 steps, while *Inv-NoCom* agents take about 5 steps. This indicates that agents with more accurate communication require extra steps to send meaningful signals, increasing their bandwidth and hence, longer episode length. Another possible strategy for implicit communication is bumping into each other; however, our analysis in subsection C.7 shows this is not the case. The similar analysis above on implicit communication in *ScoreG* is also applicable in *TemporalG* (subsection D.1). The result is consistent with prior research. (Mordatch & Abbeel, 2018) found that agents can use their physical body to send signals to others (e.g., through actions like pointing).

**Population size affects language structure [*ScoreG*-FC].** Linguistic and cognitive studies suggest that larger communities are more likely to develop compositional languages (Reali et al., 2018; Raviv et al., 2019). Motivated by this, we examine how compositionality changes with population size. The results are shown in Figure 6a. For *topsim*, XP agents exhibit a gradual decrease as population size increases, echoing prior findings in Rita et al. (2022). In contrast, XP+SP agents display a more stable upward trend in *topsim* with larger populations, suggesting that social training conditions help maintain or improve topographic structure even as group size grows. Turning to *repcom*, the new results reveal trends that differ from earlier observations. For XP agents, *repcom* now shows a slight but consistent downward trajectory as population size increases, with values slowly declining from size 2 to 15. XP+SP agents likewise exhibit a decreasing pattern. Taken together, these findings indicate that population size influences structural properties of the emergent language in distinct ways depending on the training setup. While *topsim* trends

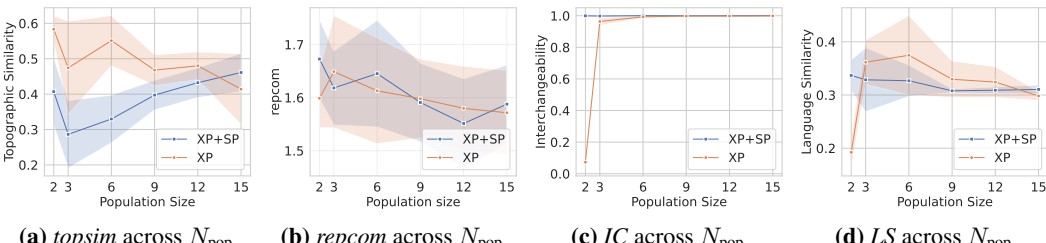

**(a)** *topsim* across $N_{\text{pop}}$    **(b)** *repcom* across $N_{\text{pop}}$    **(c)** *IC* across $N_{\text{pop}}$    **(d)** *LS* across $N_{\text{pop}}$

**Figure 6. The effect of population training with fully-connected social networks on language properties.** (a) Topographic Similarity (*topsim*), (b) Representational Compositionality (*repcom*), (c) Interchangeability (*IC*), and (d) Language Similarity (*LS*) as a function of population sizes are shown. See subsection A.3 for metrics details.

align with and extend previous studies reporting decreases (Rita et al., 2022) or stability (Michel et al., 2023), the updated *repcom* results suggest that larger populations do not necessarily enhance representational compositionality, offering a complementary perspective on how community size shapes emergent communication.

**Population size affects language similarity and interchangeability [*ScoreG*-FC].** Figure 6c and 6d show that IC and LS saturate as population size increases. XP+SP agents maintain high metrics across all $N_{\text{pop}}$, while XP agents plateau for $N_{\text{pop}} \geq 3$. This suggests that self-play and population diversity both promote shared, interchangeable communication, contrasting with findings in Kim & Oh (2021) where larger populations reduced similarity. This discrepancy may stem from their unidirectional communication and centralized training, resulting in different optimal solutions.

## 6. Discussion

Emergent language offers a useful way to explore which ecological and cognitive factors could support the development of structured communication. In this work, we examine how communication emerges in multi-agent Foraging Games under different environmental and social pressures. Our simulations provide insight into when and why agents benefit from communicating in cooperative settings. In the non-cooperative control experiment (subsection C.3), communication did not arise at all, indicating that cooperation is the pressure that makes signaling useful in this environment (Tomasello, 2010).

Importantly, the agents are not exposed to human language, yet under cooperative pressures and partial observability, they develop behaviors exhibiting several properties discussed in linguistic theory (Hockett, 1960). **Interchangeability** emerges through self-play training or social interactions within larger populations (Table 1). **Arbitrariness** is demonstrated by two cross-play agents developing non-interchangeable languages that are nonetheless mutually intelligible enough to enable successful cooperation and a high task success rate (Table 1 and Figure S4a). **Compositionality** is indicated by high *repcom* and *topsim* scores (Figure 6b, Figure 6a).

**Displacement** is demonstrated by above-chance linear decoding accuracy of items' positions, scores, and spawn times, showing that agents can refer to when and where currently invisible events happen (Figure 4a). **Cultural transmission** is evidenced by the gradual decline in LS and SR with increasing distance in structured populations, despite no direct training between distant pairs (Figure 3, Figure S5d, Figure S6d). This suggests that language is transmitted and shaped through interaction within structured social networks. We emphasize that these should be interpreted as functional analogues within the simulated environment, not as claims about human language evolution. Rather, they illustrate how certain communicative capacities can arise from ecological structure, shared intentions, and learning dynamics in multi-agent systems (Wacewicz & Żywiczyński, 2015).

**Limitations**    The language that emerges in FG is not directly comparable to natural language: it remains simple and lacks syntax, morphology, and other structural features of human communication. FG also provides limited semantic richness, as agents communicate primarily about a small set of task variables (time, location, and score). More broadly, emergent languages can show both similarities (Yao et al., 2022; Boldt & Mortensen, 2025) and clear divergences (Kottur et al., 2017; Chaabouni et al., 2019) from natural language, and it remains an open question whether they can be made substantially more human-like or interpretable. Achieving more human-like communication likely requires richer, open-ended environments with greater semantic depth than our current framework. Standard MARL methods may struggle with such complexity, requiring further advancement in the field. Furthermore, our model omits turn-taking, which remains a key challenge for future emergent communication research. Finally, the EC community lacks a universal metric for compositionality: *topsim* is widely used but not theoretically pinned to compositionality, while *repcom* is better grounded yet depends on Kolmogorov-complexity estimates that vary with model architecture. Developing metrics that are both principled and robust remains an open problem.

## Impact Statement

This work contributes foundational insights to emergent communication and multi-agent learning, with relevance to machine learning and cognitive science. The work is purely theoretical, with no immediate real-world deployment or direct societal impact.

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

# A. Implementation and Training Details

## A.1. Learning to Communicate with PPO

Agents in our setting learn to act and communicate simultaneously. As illustrated in Figure 2, each agent receives an observation from the environment and a message from its partner sent at the previous time step. Based on this input, the agent produces a discrete action and message. During training, the agent also estimates a value function to guide learning. In the following, we formalize this process using the multi-agent reinforcement learning (MARL) framework and describe how action and communication are jointly optimized via the MARL algorithm. The method most closely related to ours is RIAL (Foerster et al., 2016). However, unlike RIAL and related approaches, we do not assume parameter sharing or use a centralized critic. While Independent PPO (De Witt et al., 2020) employs parameter sharing among independent agents, a technique that can help stabilize learning (Sun et al., 2023), our method explicitly removes this parameter sharing. These design choices make our setting more reflective of human-like conditions, where individuals operate independently and learn through decentralized interaction. Moreover, by avoiding parameter sharing, our method enables the study of linguistic and behavioral heterogeneity across agents.

**Problem Formulation**  Our learning setting is a two-player variant of decentralized partially observable Markov decision processes (DecPOMDP) (Bernstein et al., 2002). A two-player DecPOMDP with communication is formally defined as the tuple $(S, \mathcal{A}_1, \mathcal{A}_2, \mathcal{M}_1, \mathcal{M}_2, \Omega_1, \Omega_2, T, O, r, \gamma, H)$, where $S$ represents the state space, $\mathcal{A} \equiv \mathcal{A}_1 \times \mathcal{A}_2$ denotes the joint-action space, $\mathcal{M} \equiv \mathcal{M}_1 \times \mathcal{M}_2$ denotes the joint-message space, $\Omega \equiv \Omega_1 \times \Omega_2$ signifies the joint-observation space (where $\Omega_1$ and $\Omega_2$ are the individual observation spaces), $T(s'|s, a_1, a_2)$ provides the transition probability from state $s$ to $s'$ upon executing the joint action $(a_1, a_2)$, $O(o_1, o_2|s)$ represents the conditional probability of observing the joint observation $(o_1, o_2)$ given the current state $s$, $r(s, a_1, a_2)$ is the shared reward function, $\gamma$ is the discount factor for rewards, and $H$ denotes the horizon length.

The first and second agents are controlled by neural policies $\psi_1 = (\pi_1, \phi_1)$ and $\psi_2 = (\pi_2, \phi_2)$, respectively, where the subscript indicates the agent identity. Here, $\pi$ denotes the action policy and $\phi$ denotes the communication policy. Each neural policy $\psi$ is parameterized by $\theta$. At each time step $t$, agents receive the joint observation $o^{(t)} = (o_1^{(t)}, o_2^{(t)})$ and the previous joint message $m^{(t-1)} = (m_1^{(t-1)}, m_2^{(t-1)})$. Each agent $i$ maintains a trajectory $\tau_i^{(t)}$ consisting of its own history up to time $t$. Conditioned on its trajectory $\tau_i^{(t)}$ and the other agent's previous message $m_j^{(t-1)}$, agent $i$ samples its action and message as follows: $a_i^{(t)} \sim \pi_i(a_i^{(t)} \mid m_j^{(t-1)}, \tau_i^{(t)})$ and $m_i^{(t)} \sim \phi_i(m_i^{(t)} \mid m_j^{(t-1)}, \tau_i^{(t)})$. The joint trajectory can be written as $\tau = (o_0, a_0, m_0, r_1, o_1, ..., o_{H-1}, a_{H-1}, m_{H-1}, r_H, o_H) \in \mathcal{T} \equiv (\Omega \times \mathcal{A} \times \mathcal{M} \times \mathbb{R})^H$. The return of a trajectory $\tau$ is defined as $G(\tau) = \sum_{t=1}^{H} \gamma^{t-1} r_t$, and the expected return of the joint policy $(\psi_1, \psi_2)$ is $J(\psi_1, \psi_2) = \mathbb{E}_{\tau \sim p(\tau|\theta_1, \theta_2)} G(\tau)$, where $p(\tau|\theta_1, \theta_2)$ denotes the distribution over trajectories induced by the joint policy.

**Learning to Act and Communicate with PPO**  As described above, we treat the communication policy the same as the action policy, i.e., we jointly optimize them with policy gradients. This is because both policies output discrete decisions based on the agent's observation history, and both can be optimized with similar objective functions under policy gradient methods. We use PPO (Schulman et al., 2017) because of its effectiveness in optimizing discrete actions, sample efficiency, and good performance in multi-agent learning (Li et al., 2023; De Witt et al., 2020; Yu et al., 2022). We first define $r_a^{(t)}(\theta) = \frac{\pi(a^{(t)}|\tau^{(t)})}{\pi_{\text{old}}(a^{(t)}|\tau^{(t)})}$ and $r_m^{(t)}(\theta) = \frac{\phi(m^{(t)}|\tau^{(t)})}{\phi_{\text{old}}(m^{(t)}|\tau^{(t)})}$ as the ratios of action policy and communication policy (Schulman et al., 2017), respectively. The objective function of the neural policy $\theta$ can be written as $\mathcal{J} = \mathcal{J}_a + \mathcal{J}_m + \mathcal{J}_{\text{ent}}$. We use Generalized Advantage Estimation (GAE) (Schulman et al., 2016) to estimate the advantage function. We use a single advantage function $\hat{A}^{(t)}$ estimated from the shared representation output by the LSTM, which is used for optimizing both the action and communication policies. Each agent is trained independently using standard single-agent PPO, which treats its partner as part of the environment, i.e., fully decentralized training. The first term optimizes the action policy $\pi$ using the estimated advantage function:

$$\mathcal{J}_a = \mathbb{E}_{t, \tau \sim p(\tau|\theta, \cdot)} \left[ \min \left( r_a^{(t)}(\theta) \hat{A}^{(t)}, \ \text{clip}(r_a^{(t)}(\theta), 1 - \epsilon, 1 + \epsilon) \hat{A}^{(t)} \right) \right]. \tag{1}$$

The second term optimizes the communication policy $\phi$ in the same manner:

$$\mathcal{J}_m = \mathbb{E}_{t, \tau \sim p(\tau|\theta, \cdot)} \left[ \min \left( r_m^{(t)}(\theta) \hat{A}^{(t)}, \ \text{clip}(r_m^{(t)}(\theta), 1 - \epsilon, 1 + \epsilon) \hat{A}^{(t)} \right) \right]. \tag{2}$$

The final term encourages exploration by maximizing the entropy of both action and message distributions:

$$\mathcal{J}_{\text{ent}} = -\mathbb{E}_{t,\tau \sim p(\tau|\theta,\cdot)} \left[ \lambda_a \sum_a \pi(a|\tau^{(t)}) \log \pi(a|\tau^{(t)}) + \lambda_m \sum_m \phi(m|\tau^{(t)}) \log \phi(m|\tau^{(t)}) \right]. \tag{3}$$

**Neural Architecture**   In Figure 2, the agent architecture is divided into input and output components. The encoders are essential for converting heterogeneous inputs such as messages, grid observations, and spatial coordinates into a unified feature representation, enabling effective downstream processing and integration by the LSTM. On the input side, the *message encoder* $\mathcal{E}_{\mathcal{M}}$, *grid encoder* $\mathcal{E}_{\mathcal{X}}$, and *position encoder* $\mathcal{E}_{\mathcal{P}}$ process the incoming message from the other agent, the grid observation, and the agent's current position, respectively. The extracted features from these encoders are concatenated and fed into an LSTM (Hochreiter & Schmidhuber, 1997), which maintains temporal memory. On the output side, the *message head* $\mathcal{D}_{\mathcal{M}}$, *action head* $\mathcal{D}_{\mathcal{A}}$, and *value head* $\mathcal{D}_{\mathcal{V}}$ generate the outgoing message, the selected action, and the estimated state value, respectively. All encoders and heads are implemented as shallow multilayer perceptrons (MLPs). The message encoder $\mathcal{E}_{\mathcal{M}}$ includes an embedding layer that maps each vocabulary index to a real-valued vector before passing it to the MLP. Our code is built on the widely used PPO-LSTM implementation by (Huang et al., 2022).

## A.2. Model Training

We train our models on a single GeForce RTX 4090. However, due to the small model size and low environment complexity, training can also be run entirely on a CPU as well. For populations smaller than 3, training takes approximately 16 hours, depending on the CPU and the number of available cores. The longest training time occurs with a population size of 15 and takes up to 3 days. All experiments are repeated with three different random seeds. We use the ADAM optimizer (Kingma & Ba, 2015) with an initial learning rate of 0.00025, which linearly decays over time. We report the hyperparameters of the architecture and training algorithm in Table S1 and Table S2. The examples of learning dynamics for *ScoreG* and *TemporalG* are shown in Figure S1 and Figure S2.

**Table S1.** Neural architecture hyperparameters.

| Component | Hyperparameter |
|---|---|
| Visual encoder | 4-layer MLP: [256, 256, 128, 16] |
| Visual input shape | $(1, 5, 5)$ grayscale grid |
| Position encoder | Linear(2, 4) |
| Message embedding size | 16 |
| Message encoder | Embedding(Vocab. Size, 16) + Linear(16, 16) |
| LSTM hidden size | 128 |
| LSTM input size | 16 (visual) + 4 (position) + 16 (message) = 36 |
| Action head | Linear(128, `num_actions`) |
| Value head | Linear(128, 1) |
| Message head | Linear(128, 16) |
| Weight initialization | Orthogonal (std = $\sqrt{2}$), output layers: std = 0.01 |

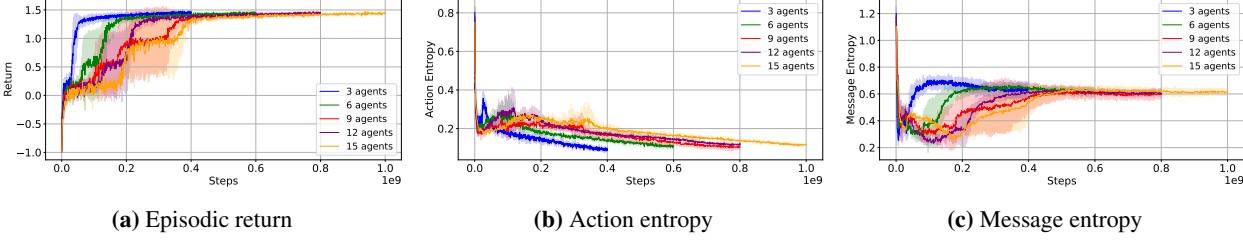

**(a)** Episodic return      **(b)** Action entropy      **(c)** Message entropy

**Figure S1. Larger population takes longer time to converge.** The figure shows learning dynamics of agents with different population sizes in *ScoreG*. Return, action entropy, and message entropy are plotted over environment steps, averaged across three seeds. A single step is defined as a single interaction with the environment.

**Table S2.** PPO hyperparameters.

| Hyperparameter | Value |
|---|---|
| Total time steps | $2 \times 10^9$ |
| Learning rate | $2.5 \times 10^{-4}$ |
| Number of environments | 128 |
| Number of steps per rollout | 32 |
| Number of minibatches | 4 |
| Number of update epochs | 4 |
| Discount factor $\gamma$ | 0.99 |
| GAE $\lambda$ | 0.95 |
| Clip coefficient | 0.1 |
| Clip value loss | True |
| Normalize advantages | True |
| Action entropy coefficient | 0.01 |
| Message entropy coefficient | 0.002 |
| Value function coefficient | 0.5 |
| Max gradient norm | 0.5 |
| Learning rate annealing | True |
| Target KL divergence | None |
| Batch size | 4096 |
| Minibatch size | 1024 |
| Optimizer | Adam ($\epsilon = 10^{-5}$) |

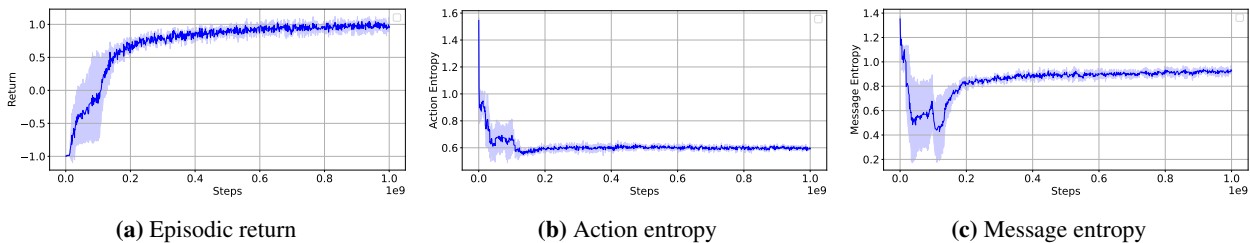

(a) Episodic return  (b) Action entropy  (c) Message entropy

**Figure S2. Learning dynamics of three XP agents in *TemporalG*.** Agents start to converge at around one billion environment steps.

### A.3. Metrics

**Topographic Similarity (topsim)**   Topographic similarity measures the alignment between the structure of the message space and the semantic space (e.g., environmental states or properties). It is defined as the Spearman rank correlation between all pairwise distances in these two spaces (Brighton & Kirby, 2006; Lazaridou et al., 2018). A higher *topsim* indicates a more compositional and consistent mapping between semantic meanings and messages. We define the semantic space using ground-truth attributes of the environment, specifically the scores and positions of items in the *ScoreG*. To implement the topographic similarity metric, we use the EGG toolkit (Kharitonov et al., 2019).

Let $M = \{m_1, m_2, \ldots, m_N\}$ be the set of message sequences, and $S = \{s_1, s_2, \ldots, s_N\}$ be the corresponding semantic meanings. Let $\delta_M(m_i, m_j)$ and $\delta_S(s_i, s_j)$ denote the pairwise distance functions in the message and semantic spaces, respectively. Then:

$$\text{topsim} = \text{Spearman}\left(\{\delta_M(m_i, m_j)\}_{i<j}, \{\delta_S(s_i, s_j)\}_{i<j}\right) \quad (4)$$

A high topographic similarity indicates that similar messages correspond to similar semantic meanings, reflecting a structured and grounded communication protocol. We use items' positions and scores as semantic meanings.

**Calculate Semantic Meaning Vectors:** The semantic space is constructed from each item's spatial position and score. We

represent the semantic state of item $i$ as

$$S_i = [x_i, y_i, s_i],$$

where $(x_i, y_i)$ denotes the item's location and $s_i$ denotes its score. These semantic vectors are used to compute pairwise distances, which are then compared with pairwise message distances when calculating *topsim*.

**Representational Compositionality (repcom)** quantifies how well semantic representations can be reconstructed from messages using a low-complexity mapping function (Elmoznino et al., 2025). The metric is grounded in algorithmic information theory and is defined as a compression ratio that captures both the expressivity of the semantic space and the simplicity of the mapping from messages to meanings.

Let $S \in \mathbb{R}^{N \times D}$ be the matrix of semantic representations (one vector $s_i$ per item), and let $M = \{m_1, m_2, \ldots, m_N\}$ be the corresponding discrete message sequences produced by the agents.

In the theoretical formulation, representational compositionality is defined as:

$$\text{repcom} = \frac{K(S)}{K(f) + K(S \mid M, f)}, \tag{5}$$

where $f$ denotes the semantics function mapping messages to predicted semantic vectors, $K(f)$ is the Kolmogorov complexity of this mapping (its description length), and $K(S \mid M, f)$ is the number of bits required to encode $S$ given that they are generated from messages via $f$. A language achieves high *repcom* when a *simple* mapping function $f$ (low $K(f)$) can *accurately* reconstruct semantic vectors (low $K(S \mid M, f)$), indicating a systematic and compositional relationship between messages and meanings.

Since Kolmogorov complexities are not computable, we follow the empirical estimation procedure of Elmoznino et al. (2025), in which $K(S \mid M, f)$ is approximated using the model's prediction loss on semantic vectors. Given a parametric predictor $f_\theta$ trained to map each message $m_i$ to its semantic meaning $s_i$, we estimate:

$$\widehat{K(S \mid M)} \approx \sum_{i=1}^{N} \mathcal{L}\big(s_i, f_\theta(m_i)\big), \tag{6}$$

where $\mathcal{L}$ is a reconstruction loss (e.g., mean squared error). Similarly, the complexity of the mapping $K(f)$ is approximated implicitly through the inductive bias and capacity of the predictor $f_\theta$; simpler models correspond to lower effective complexity. In practice, $\mathcal{L}$ is calculated using prequential coding (Blier & Ollivier, 2018).

A higher *repcom* indicates that semantic representations can be generated from messages through a low-complexity, high-accuracy mapping, reflecting a structured and compositional communication protocol. In contrast, if reconstructing $S$ requires a complex or arbitrary mapping, or if reconstruction is inaccurate, *repcom* decreases.

**Language Similarity (LS)** quantifies the token-level similarity between two agents based on their communication over multiple episodes, starting from the same initial condition. A higher score means the agents tend to use the same messages in similar contexts, indicating stronger convergence in their communication strategies. For a given episode $e$, the agents $i$ and $j$ produce their respective message sequences $M_i^{(e)}$ and $M_j^{(e)}$. Let $\mathcal{D}_{\text{edit}}(\cdot)$ represent the normalized edit distance function, and let $N_e$ denote the total number of evaluation episodes. The Language Similarity (LS) between agents $i$ and $j$ is defined as:

$$\text{LS(i,j)} = \frac{1}{N_e} \sum_{e=1}^{N_e} 1 - \mathcal{D}_{\text{edit}}(M_i^{(e)}, M_j^{(e)}) \tag{7}$$

We can then compute the average LS across all pairs of agents in the entire population as follows: $\text{LS} = \frac{1}{N_{\text{pop}}(N_{\text{pop}}-1)} \sum_{i=1}^{N_{\text{pop}}} \sum_{j \neq i}^{N_{\text{pop}}} \text{LS}(i,j)$.

**Interchangeability (IC)** refers to a property of language wherein a speaker can both send and understand the same linguistic signals (Hockett, 1960). In the context of agents, this means that an agent should understand the language it produces. To evaluate interchangeability in agents, we embed the same neural network in two different agent bodies and

assess their performance in the game. We compare an agent's success when paired with itself to its success when paired with other agents. Formally, consider a set of $N_{\text{pop}}$ agents. Let $\text{SR}(i, j)$ denote the success rate of an agent $i$ when playing with an agent $j$. Therefore, we propose that interchangeability (IC) can be defined as:

$$\text{IC} = (N_{\text{pop}} - 1) \times \frac{\sum_{i=1}^{N_{\text{pop}}} \text{SR}(i, i)}{\sum_{i=1}^{N_{\text{pop}}} \sum_{j \neq i}^{N_{\text{pop}}} \text{SR}(i, j)} \tag{8}$$

## B. Additional Experiments and Results on Cultural Transmission

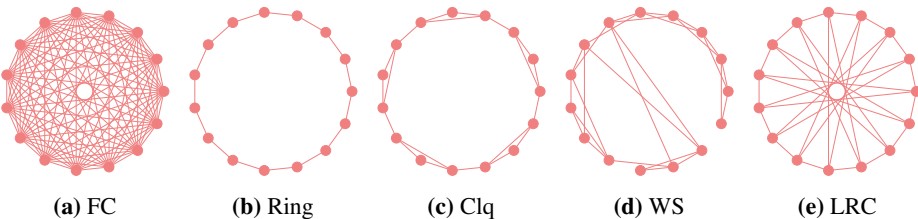

|  (a) FC | (b) Ring | (c) Clq | (d) WS | (e) LRC |

**Figure S3. Social network structures:** Nodes represent agents, and edges denote pairs of agents that are co-trained. Network structures are abbreviated as follows: **FC – Fully Connected**, **Ring – Ring Structure**, **Clq – Ring with Cliques**, **WS – Watts-Strogatz**, and **LRC – Ring with Long-Range Connections**

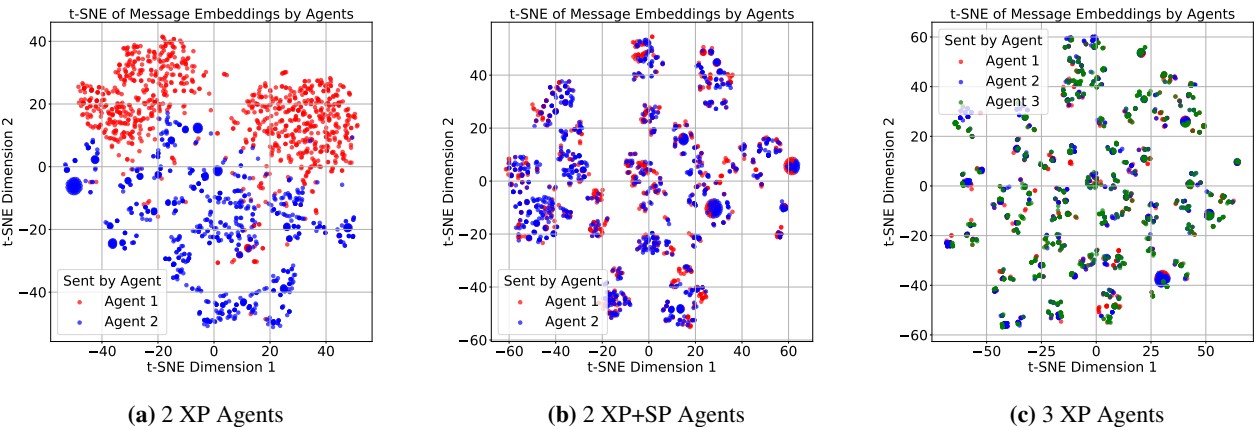

**(a)** 2 XP Agents  **(b)** 2 XP+SP Agents  **(c)** 3 XP Agents

**Figure S4. Message embeddings under different training regimes.** In the t-SNE space of embedding vectors, sequences of messages from two agents are clearly separable in the 2-XP-Agents setting. (a) Messages from different agents occupy distinct regions of the embedding space (indicating different languages). (b-c) Messages from different agents largely overlap (indicating similar languages).

We further examine whether adding additional connections, while keeping social networks sparse, can enhance cultural transmission. To this end, we employ two networks with small-world properties, as illustrated in Figure S3. The first is the Watts-Strogatz (WS) network, constructed with parameters $k = 4$ (neighbors) and $\beta = 0.2$. The second is a ring structure with long-range connections (LRC) network, built upon a standard ring lattice with additional long-range edges. Specifically, the LRC network is generated by iteratively linking the farthest nodes on the ring lattice, serving as a simple proxy for reducing average path length and thereby producing a more small-world-like structure.

**More cliques and long-range connections both enhance cultural transmission [ScoreG-P15-XP]** We examine the impact of adding cliques or long-range connections to the ring-structured network on language transmission. Results are shown in Figure S7a and Figure S7b. For WS and LRC, both LS and SR decrease slightly with social distance, demonstrating successful language transmission to non-co-trained agents in the population. For Clq, LS at distance 1 is already lower and declines more steeply with distance than the others. SR also falls sharply, dropping from $0.95$ at distance 1 to $0.55$ at distance 5. This result is intuitive because Clq is constructed by adding only five connections to the ring network, which may not be enough for the population to form similar and compatible languages, whereas the others are constructed by adding many more connections to the ring network.

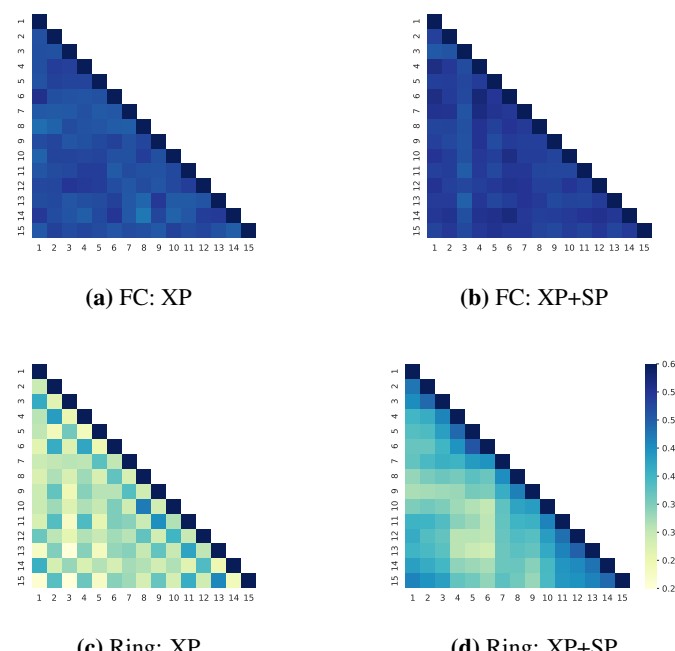

**(a)** FC: XP      **(b)** FC: XP+SP

**(c)** Ring: XP      **(d)** Ring: XP+SP

**Figure S5. Language Similarity (LS) between agent pairs under different social structures and training strategies.** (a–b) Under the fully connected (FC) social network, both training regimes (XP and XP+SP) yield similar languages across agents. (c) With XP training in the Ring network, agents develop different languages from their partners (evident from the color switching between light and dark along the axis perpendicular to the diagonal). (d) With XP+SP training in the Ring network, agents and their partners converge to similar languages.

## C. ScoreG: Additional Results and Analyses

### C.1. Ablation: Disembodiment (Referential Game)

In this variant, agents are immobilized and remain fixed near their respective items. The game structure and goal are the same as those in *ScoreG*, but without requiring agents to move to pick up the target item. Specifically, the action space is restricted to {*select my item*, *select another item*, *idle*}. The game succeeds only if both agents correctly select the target item at the final timestep ($t = 6$). Any premature selection causes immediate failure. Rewards are binary: $+1$ for success and $-1$ for failure, with no advantage for finishing early since the episode length is fixed.

As shown in Figure S8, agents exchange messages conveying *item score* information, but not spatial information, a result confirmed by chance-level decoding accuracy. Each agent still observes its own static position, which in principle allows item locations to be inferred. However, this positional information is task-irrelevant, as agents cannot move.

### C.2. Ablation: Unidirectional Communication

We study a setting where only one agent (the sender) can transmit messages, while the other (the receiver) cannot send messages back—i.e., the sender receives no information from its partner. Two of the three agents are randomly selected and assigned as sender or receiver, and XP training is used. This setup reflects the unidirectionality typical of referential games.

Agents are evaluated under two conditions: *Normal* (scores sampled from $[2, 4, 6, \ldots, 248]$) and *Hard* (scores sampled from $[160, 162, \ldots, 240]$). The result is shown in Table S3. In the *Normal* condition, agents achieve a high SR of $0.829$ because certain score configurations allow effective guessing. For instance, if the sender observes a low score of 32 and the receiver observes 232, the receiver infers that the sender sees 32 and can wait, while the sender correctly judges that its item is unlikely to be a target and searches for the other.

In the *Hard* condition, however, SR drops to chance level ($0.508$). Because all scores are moderate, agents cannot reliably infer whether their item is a target. This result underscores the importance of bidirectional communication for achieving high SR in the ScoreG game.

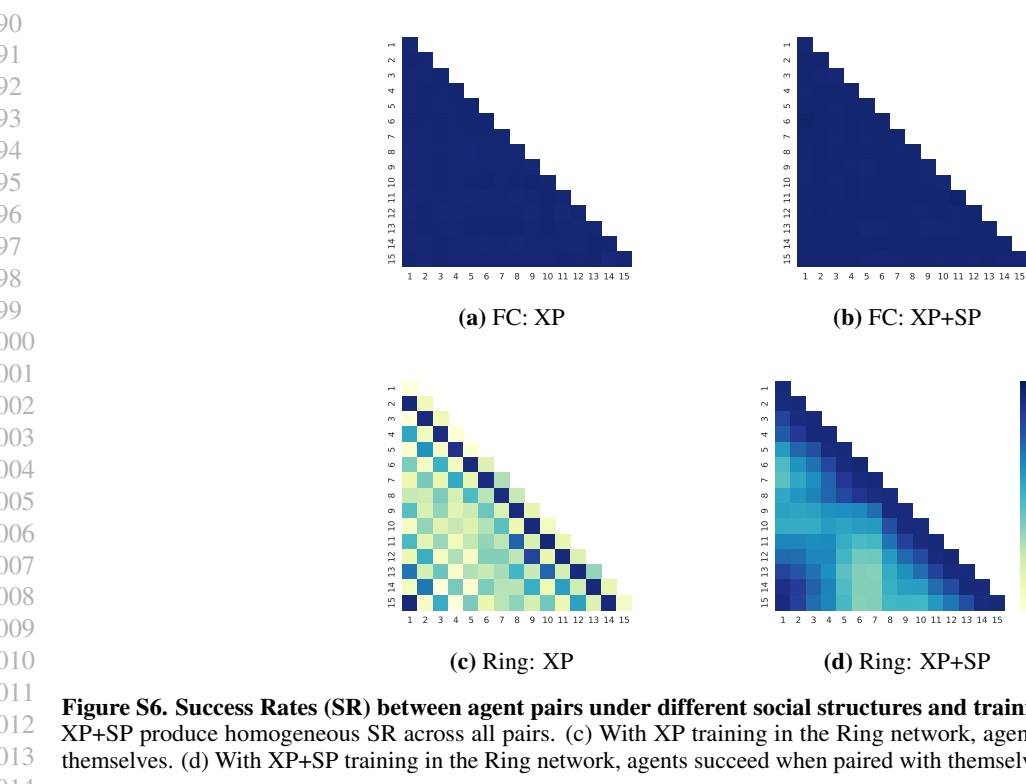

**(a)** FC: XP  **(b)** FC: XP+SP

**(c)** Ring: XP  **(d)** Ring: XP+SP

**Figure S6. Success Rates (SR) between agent pairs under different social structures and training strategies.** (a–b) Both XP and XP+SP produce homogeneous SR across all pairs. (c) With XP training in the Ring network, agents fail when paired with copies of themselves. (d) With XP+SP training in the Ring network, agents succeed when paired with themselves.

**Table S3.** Performance of three XP agents with **unidirectional communication** under different test conditions.

| Test Condition | SR | Length |
|---|---|---|
| Normal | $0.829 \pm 0.022$ | $5.166 \pm 0.162$ |
| Hard | $0.508 \pm 0.011$ | $5.934 \pm 0.503$ |

## C.3. Ablation: No Cooperation

**Table S4.** Ablation study on non-cooperative tasks (*IndividualG*). We report average rewards for two agents and the average episode length (mean $\pm$ standard deviation) under three communication conditions.

| Condition | Avg. Reward (First Player) | Avg. Reward (Second Player) | Avg. Length |
|---|---|---|---|
| *Normal* | $1.770 \pm 0.180$ | $1.878 \pm 0.194$ | $13.762 \pm 1.240$ |
| *Ablate-Noise* | $1.822 \pm 0.152$ | $1.910 \pm 0.171$ | $13.300 \pm 1.364$ |
| *Ablate-Zero* | $1.740 \pm 0.135$ | $1.945 \pm 0.179$ | $13.148 \pm 1.124$ |

A leading hypothesis suggests that language, or communication more broadly, emerges because agents need to share their intentions or knowledge to cooperate effectively. To test this hypothesis, we design an additional experiment in which agents are embodied in the same environment but pursue individual goals (i.e., each agent separately picks up any items in the map). In this setting, there is no mutual incentive for agents to pick up the same item—in other words, shared intentionality is removed. We use the same neural network architecture and training regime (XP) as in the *ScoreG* experiments. Agents can still send and receive messages from their partners, but we hypothesize that communication will be less useful because coordination is not required to achieve shared rewards. We describe this environment, called *IndividualG*, below.

**IndividualG**   The game structure is similar to *ScoreG*, with agents placed in a $6 \times 6$ grid world and sharing the same action space. However, there are several key differences. First, no scores are assigned to items, and each item requires only a single agent to pick it up. Consequently, no shared reward is available to cooperative agents, and communication is unnecessary to succeed. Second, an agent receives a reward of +1 for each item it successfully collects. Items cannot be picked up by two

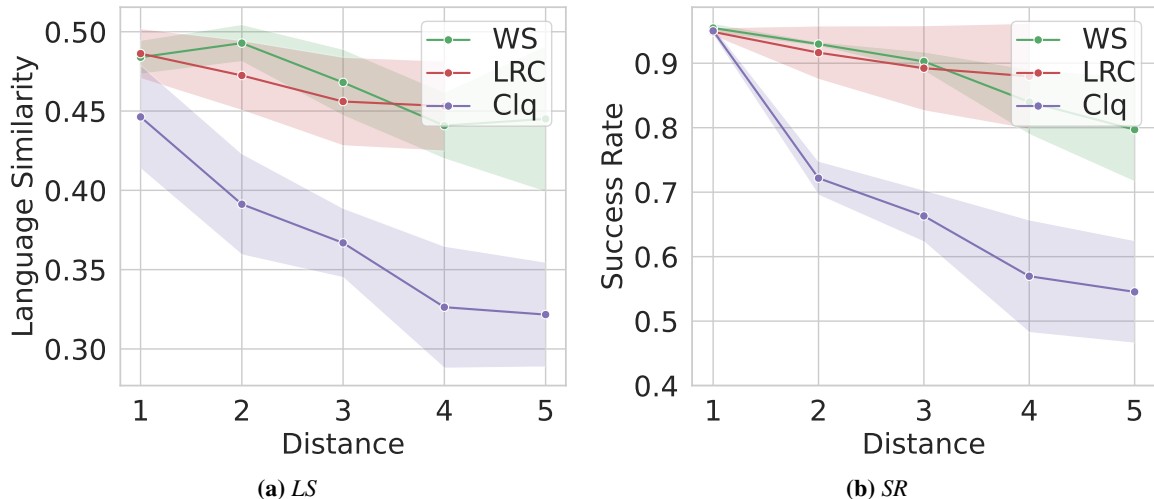

(a) *LS*  (b) *SR*

**Figure S7. More cliques and long-range connections both enhance cultural transmission** ($N_{\text{pop}} = 15$)**:** We plot *LS* and *SR* (See subsection A.3) as a function of the shortest-path distance between two agents. A distance of 1 indicates that two agents were co-trained, while a distance greater than 1 indicates that the agents were never paired during training. Note that the maximum shortest path of LRC is only 4 while other cases are 5.

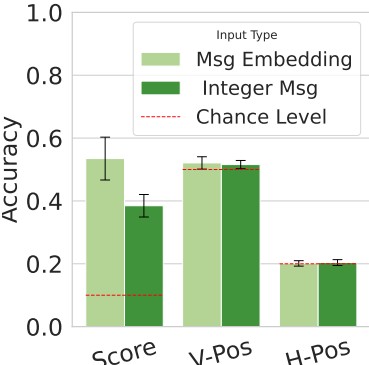

**Figure S8. Decoding item states from messages produced in a no-body referential game.** *V-Pos/H-Pos* are item positions, and *Score* is item value. *Integer Msg* uses raw message chains composed of integer sequences. *Msg Embedding* uses chained embeddings mapped from a lookup table. Error bars show standard deviations.

agents simultaneously. There are four items in total, so the maximum reward for a single agent is 4. The Nash equilibrium is for each agent to pick up two items. The game ends after 20 time steps or once all items have been collected.

**Results**    The goal of this experiment is to test whether communication is useful in non-cooperative tasks. In other words, can meaningful communication still emerge when cooperation is unnecessary? We train three XP agents and evaluate them under three conditions: *Normal*, *Ablate-Zero*, and *Ablate-Noise*. In *Ablate-Zero*, agents' messages are replaced with zero tokens, preventing the transmission of meaningful signals. In *Ablate-Noise*, agents' messages are replaced with random samples drawn from a uniform distribution instead of the policy's output distribution. This setup forces agents to misinterpret their partners' messages if the communication learned during training had any effect. In each episode, two of the three agents are randomly assigned as the **First Player** and **Second Player**. We report the average reward (corresponding to the average number of items collected per episode) for both players to test for potential asymmetries in their behavior. Results are shown in Table S4. Across all three conditions, agents achieve similar performance, with no observable difference between the first and second players. This suggests that communication does not provide a measurable advantage when agents pursue independent rewards. In other words, without shared goals or incentives for coordination, the environment does not pressure agents to develop or rely on meaningful communication.

## C.4. Analysis: Varying Vocabulary Size

In *ScoreG*, the agents typically complete the game in approximately 5-6 time steps because we introduce time pressure to encourage agents to finish the game as soon as possible. Varying the vocabulary size alters the capacity of information that can be conveyed through the message sequence. Here we train XP agents with $N_{\text{pop}} = 3$ with different vocabulary sizes. The default vocabulary size we used in the main text was 4 (i.e., $\mathbf{m_t} \in \{0, 1, 2, 3\}$). A smaller vocabulary should, in principle, encourage greater compositionality due to a smaller capacity.

As shown in Table S5, *topsim* remains relatively stable across vocabulary sizes of 4, 8, and 16. However, when the vocabulary size increases to 32, the *topsim* score drops to $0.25$.

The *repcom* scores, in contrast, do not follow this pattern. Although they lie within a narrow range, their means show a slight downward trend: the highest value occurs at a vocabulary size of 4 ($1.692$), and the lowest at 32 ($1.566$). Despite this monotonic decrease, the differences are small and the standard deviations overlap, indicating that *repcom* does not strongly align with the changes observed in *topsim*.

**Table S5. Ablation Study on Vocabulary Size.** Effect of vocabulary size on communication and performance metrics. Values are reported as mean $\pm$ standard deviation.

| Vocab. Size | topsim | repcom | IC | Self-SR | Cross-SR |
|:---:|:---:|:---:|:---:|:---:|:---:|
| 4 | $0.311 \pm 0.111$ | $1.692 \pm 0.126$ | $0.966 \pm 0.017$ | $0.939 \pm 0.021$ | $0.972 \pm 0.004$ |
| 8 | $0.352 \pm 0.032$ | $1.586 \pm 0.084$ | $0.902 \pm 0.056$ | $0.880 \pm 0.057$ | $0.975 \pm 0.003$ |
| 16 | $0.309 \pm 0.052$ | $1.634 \pm 0.089$ | $0.954 \pm 0.030$ | $0.933 \pm 0.030$ | $0.978 \pm 0.001$ |
| 32 | $0.251 \pm 0.034$ | $1.566 \pm 0.075$ | $0.954 \pm 0.031$ | $0.931 \pm 0.031$ | $0.977 \pm 0.001$ |

## C.5. Generalization to Unseen Positions

We study how well communication generalizes when it comes to unseen object positions. We trained three XP agents to perform the ScoreG task on 17 food locations in a 5×5 grid and evaluated their generalization to 8 unseen locations. Train and test food locations are randomly selected.

**Emergent language generalizes to unseen spatial configurations.** We ask whether the emergent language developed by the agents supports reference to positions that were never observed during training. If messages encode spatial information in a systematic way, a linear classifier should be able to decode unseen positions from message embeddings. Following the decoding analysis in the main text, we train linear models to predict each component of the item state—score, vertical position, and horizontal position. As shown in Figure S9, all three components can be decoded with above-chance accuracy from the agents' messages. Moreover, message embeddings are more linearly decodable than the raw token sequences, suggesting that the learned communication protocol captures spatial structure in a manner that supports extrapolation.

**Agent coordination degrades under unseen spatial configurations.** Despite the decodability results, coordination is noticeably weaker when agents face positions they have not previously encountered. As shown in Table S6, agents with communication achieve a success rate of $0.953$ on seen locations, but performance drops to $0.756$ on unseen locations. This indicates that, while the emergent language captures some spatial regularities, the resulting behavior does not fully generalize to novel configurations and relies partly on the distribution of positions observed during training.

Our hypothesis is that the degradation in performance comes from a non-robust coordination strategy rather than from an emergent language that fails to encode the relevant information. To isolate this, we train three XP agents that have full access to all item scores and to each other's observations but are not allowed to communicate, essentially a standard MARL setting without communication. The agents are trained on 17 item locations and tested on 8 unseen locations. As shown in Table S7, their success rate drops from $0.981$ to $0.823$, indicating that the generalization gap arises largely from the inherent limits of the MARL algorithm rather than from communication alone. Improving multi-agent generalization to unseen spatial configurations is an important challenge for both MARL and emergent communication research, but addressing it is beyond the scope of the present work.

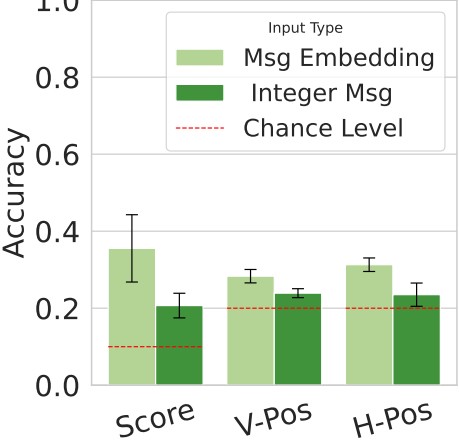

**Figure S9. Decoding unseen item states from messages.** *V-Pos/H-Pos* are item positions and *Score* is item value. *Integer Msg* uses raw message chains composed of integer sequences. *Msg Embedding* uses chained embeddings mapped from a lookup table. Error bars show standard deviations.

**Table S6.** Performance of three XP agents with communication on seen vs unseen locations.

| Location | SR | Length |
|---|---|---|
| Seen | $0.953 \pm 0.017$ | $5.197 \pm 0.099$ |
| Unseen | $0.756 \pm 0.022$ | $5.154 \pm 0.229$ |

## C.6. Analysis of Below-Chance Success Rates in Implicit Communication

In *Inv-NoCom* setting, the success rate (SR) falls below the chance level (See Figure 5a), which indicates a systematic coordination failure rather than random performance. To better understand this phenomenon, we conduct two analyses: (i) ablation of the speed reward, and (ii) categorization of failure outcomes.

**Ablation of the speed reward.** The speed reward encourages efficient task completion. Removing this reward does not substantially change SR, which remains below chance. However, the absence of the speed reward increases the average episode length (Table S8). This result shows that the speed reward primarily affects efficiency but does not resolve the core coordination issue that leads to low SR.

**Failure outcome statistics.** To diagnose why agents fail, we categorize episode outcomes into three groups: *correct pickups*, *wrong pickups*, and *no pickups*. Table S9 shows the distribution. Failures are dominated by wrong pickups (34.8%), followed by cases where agents never pick up the same item (27.4%). Together, these account for the below-chance success rate and show that agents struggle to coordinate reliably without communication.

Overall, these findings show that below-chance SR in *Inv-NoCom* setting does not result from the speed reward but from miscoordination, most prominently through wrong item pickups.

## C.7. ScoreG: Bumping and Cell Occupancy

Each grid cell can be occupied by at most one entity (either an agent or an item) at any given time. If an agent attempts to move into a cell that is already occupied by another agent, the move is blocked and the agent remains in its original position. We refer to such blocked moves as *bumping events*.

Although bumping could theoretically provide a minimal form of implicit communication (e.g., signaling another agent's location), it constitutes only a very low-bandwidth channel and becomes difficult to coordinate once agents are spatially separated. In practice, bumping is rare and therefore does not play a substantive role in coordination or communication.

**Table S7.** Performance of three XP agents without communication on seen vs unseen locations.

| Location | SR | Length |
|---|---|---|
| Seen | $0.981 \pm 0.006$ | $4.389 \pm 0.038$ |
| Unseen | $0.823 \pm 0.023$ | $4.092 \pm 0.062$ |

**Table S8.** Effect of the speed reward on success rate (SR) and average episode length.

| Reward Condition | SR | Length |
|---|---|---|
| With speed reward | 0.438 | 5.607 |
| Without speed reward | 0.446 | 7.950 |

Table S10 summarizes the average number of bumps observed per episode across different conditions.

These results confirm that while bumping exists as an artifact of the grid-world dynamics, it is too infrequent ($< 1$ bump per episode) to support reliable strategies or emergent communication.

### C.8. Positional Disentanglement

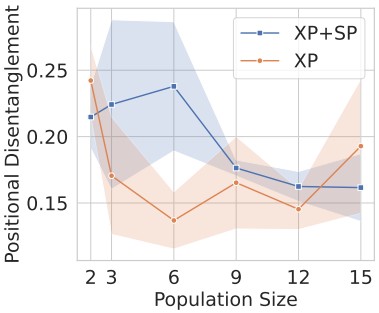

**Figure S10.** Positional Disentanglement *posdis* as a function of population size.

Topographic Similarity (*topsim*), while the most widely used metric in emergent language literature, is not the perfect measure for compositionality. (Chaabouni et al., 2020) introduced an alternative approach, measuring compositionality using disentanglement scores (Higgins et al., 2017). Specifically, **Positional Disentanglement (*posdis*)** (Chaabouni et al., 2020) quantifies how uniquely each message position encodes a single attribute, measuring the mutual-information gap between its most and second most informative attributes. *posdis* could be used to measure another aspect of compositionality in our environment because this measure requires attributes to be discrete variables.

We conduct an experiment examining the relatioship between *posdis* and population size. Data for calculating *posdis* is collected and processed in the same way as *topsim*. Specifically, the message vector consists of a sequence of integers (tokens) and the attribute vector contains the item's score and position observed by a particular agent (See subsection A.3). The result is shown in Figure S10. In contrast to *topsim* trend for XP+SP agents, *posdis* decreases with population size. We observe a parabola-like curve for XP agents, peaking at sizes 3 and 15.

It is important to note that low *posdis* does not necessarily imply low overall language compositionality. This is because *posdis* is a very strict form of compositionality where a position in a message uniquely refers to a particular attribute. This condition does not hold if we use the position of the character level. The stringency of *posdis* condition limits its general applicability.

**Table S9.** Distribution of episode outcomes without the speed reward.

| Outcome | Percentage |
|---------|------------|
| Success: Correct Pickup | 37.8% |
| Failed: Wrong Pickup | 34.8% |
| Failed: No Pickup | 27.4% |

**Table S10.** Average bumps per episode.

| Condition | Success | Failed |
|-----------|---------|--------|
| Inv-Com | 0.179 | 0.065 |
| Inv-NoCom | 0.043 | 0.057 |
| Vis-NoCom | 0.015 | 0.020 |

# D. TemporalG: Additional Results and Analyses

## D.1. Implicit Communication

As shown in Table S12, three XP agents trained without communication under the invisible-partner condition (*Inv-NoCom*) still achieved relatively high success at a fixed duration of 6, likely by adopting a waiting strategy—that is, delaying movement to infer the spawn order. Their longer episode lengths, compared to the other two conditions (*Inv-Com* and *Vis-NoCom*), support this interpretation. However, when the fixed duration was increased, the number of possible spawn times doubled (from 6 to 12), rendering the waiting strategy ineffective and causing the success rate of *Inv-NoCom* agents to fall below chance level. Similar to what was observed in *ScoreG* experiment in section 5, *Vis-NoCom* agents achieved performance comparable to *Inv-Com*, suggesting that they exploited visual cues for implicit coordination, especially under the limited set of six possible spawn times. In the *Inv-Com-Noise* condition, agents were trained with explicit communication but tested without it (messages were always zero). The low SR observed in this condition implies that meaningful messages are necessary for agents trained with communication (*Inv-Com*) to succeed at the task.

**Table S11.** Performance of XP agents in the $5 \times 5$ *TemporalG* environment with the fixed duration of 6. $N_{pop}$: population size. LS: language similarity; Cross-SR: success rate with other agents; Self-SR: success rate in self-play. All values are reported as mean ± standard deviation across held-out agent pairs.

| Model | $N_{pop}$ | LS | Cross-SR | Self-SR |
|-------|-----------|-----|----------|---------|
| XP | 3 | $0.336 \pm 0.012$ | $0.975 \pm 0.003$ | $0.962 \pm 0.006$ |

**Table S12.** Implicit communication in the $8 \times 8$ *TemporalG* with fixed durations of 6 and 12. Length denotes the average length of successful episodes (not applicable in the bottom-right cell, where some runs yielded zero successes). Abbreviations are the same as those reported in Figure 5.

| Condition | Fixed Duration = 6 | | Fixed Duration = 12 | |
|-----------|-----|--------|-----|--------|
| | SR | Length | SR | Length |
| *Inv-Com* | $0.980 \pm 0.035$ | $23.990 \pm 0.618$ | $0.961 \pm 0.021$ | $30.023 \pm 0.775$ |
| *Inv-NoCom* | $0.728 \pm 0.155$ | $28.716 \pm 2.341$ | $0.419 \pm 0.137$ | $31.738 \pm 1.780$ |
| *Vis-NoCom* | $0.939 \pm 0.093$ | $24.516 \pm 0.307$ | $0.862 \pm 0.110$ | $30.546 \pm 0.406$ |
| *Inv-Com-Noise* | $0.313 \pm 0.040$ | $28.666 \pm 0.774$ | $0.003 \pm 0.001$ | — |

### D.2. Rendezvous Behavior Analysis

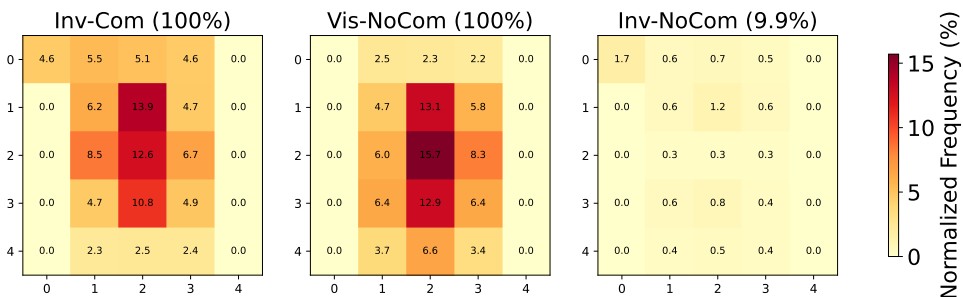

**Figure S11.** Rendezvous maps of first meeting points across conditions. Values represent normalized frequencies of encounters on the $5 \times 5$ grid. *Inv-Com* and *Vis-NoCom* agents consistently meet and converge near the center, while *Inv-NoCom* agents rarely meet and show no strong central bias.

Restricting the communication range forces agents to meet in physical proximity before they can exchange information. This environmental constraint gives rise to rendezvous points: locations where agents repeatedly converge to initiate communication. Despite this restriction, displacement remains necessary. Agents must explore the grid to observe items, store these observations in memory, and then communicate the information once they encounter their partner. Since neither agent can directly infer when all items have appeared without exchanging messages, coordination still relies on communication.

To analyze where meetings occur, we compute a rendezvous map that records the normalized frequency of the first meeting location across episodes. The resulting heatmaps reveal a pronounced central bias: agents overwhelmingly converge near the middle of the $5 \times 5$ grid (See Figure S11). This bias emerges because the center minimizes travel distance and maximizes the probability of encounter under partial observability.

When comparing conditions, clear differences emerge. *Inv-Com* agents always meet (100% of episodes) and almost exclusively converge in the central cells, reflecting highly coordinated rendezvous behavior. *Vis-NoCom* agents also meet in every episode, showing a similar central bias, suggesting that visibility supports coordination even without explicit messages. By contrast, *Inv-NoCom* agents succeed in meeting in only 9.9% of episodes, and when they do meet, the rendezvous points are weakly defined and scattered across the grid.

# E. PPO with Recurrent Memory Transformer

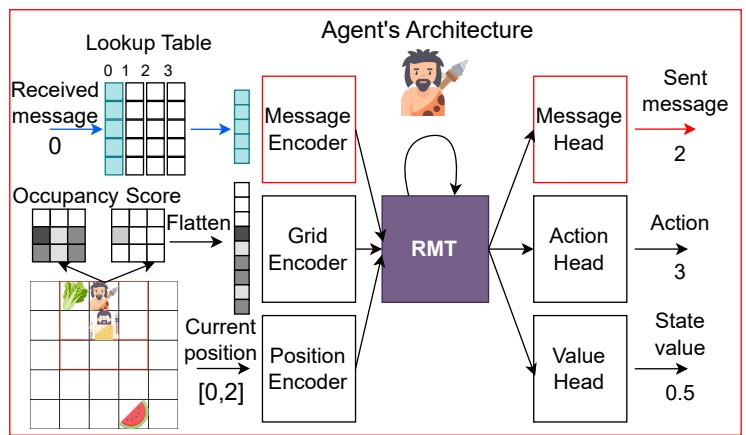 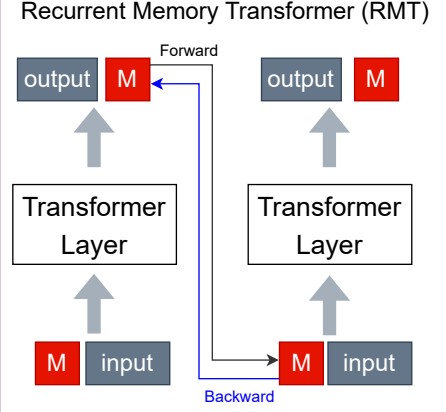

**Figure S12. PPO-RMT Architecture:** RMT can serve as a direct replacement for a traditional RNN in standard RL architectures, since it fulfills the same recurrent role. The memory token (M) functions analogously to an RNN's hidden state and provides a mechanism for backpropagation through time (BPTT).

## E.1. Architecture

The agent uses the same input encoders as in the original architecture Figure S12), but replaces the LSTM core with a tiny GPT-style Recurrent Memory Transformer (RMT) (Bulatov et al., 2022; 2024) that operates over a pair of tokens: a memory token and a current observation token. As before, the encoders map heterogeneous inputs into a common feature space, while the output heads map the resulting latent representation to the outgoing message, action, and value prediction.

On the input side, we retain the message encoder $\mathcal{E}_{\mathcal{M}}$, grid encoder $\mathcal{E}_{\mathcal{X}}$, and position encoder $\mathcal{E}_{\mathcal{P}}$. Given at time step $t$ the incoming message $m_t$, the grid observation $x_t$, and the agent's position $p_t$, the encoders produce feature vectors

$$e_t^{\mathcal{M}} = \mathcal{E}_{\mathcal{M}}(m_t) \in \mathbb{R}^{d_{\mathcal{M}}}, \qquad e_t^{\mathcal{X}} = \mathcal{E}_{\mathcal{X}}(x_t) \in \mathbb{R}^{d_{\mathcal{X}}}, \qquad e_t^{\mathcal{P}} = \mathcal{E}_{\mathcal{P}}(p_t) \in \mathbb{R}^{d_{\mathcal{P}}}.$$

All encoders are implemented as shallow MLPs; $\mathcal{E}_{\mathcal{M}}$ includes an embedding layer that maps each vocabulary index to a dense vector before applying the MLP. These features are concatenated,

$$z_t = [\, e_t^{\mathcal{X}};\, e_t^{\mathcal{P}};\, e_t^{\mathcal{M}} \,] \in \mathbb{R}^{d_{\text{in}}},$$

and projected into the transformer model dimension $d_{\text{model}}$ using a learned linear projection,

$$o_t = W_{\text{in}} z_t + b_{\text{in}} \in \mathbb{R}^{d_{\text{model}}}.$$

The agent maintains a recurrent memory state $h_{t-1} \in \mathbb{R}^{d_{\text{model}}}$, initialized to zero at the beginning of each episode. Instead of feeding $(z_t, h_{t-1})$ into an LSTM, the PPO-RMT constructs two tokens:

$$m_t = W_{\text{mem}} h_{t-1} + b_{\text{mem}} \in \mathbb{R}^{d_{\text{model}}}, \qquad o_t \in \mathbb{R}^{d_{\text{model}}},$$

and stacks them into a length-2 sequence,

$$X_t^{(0)} = \begin{bmatrix} m_t \\ o_t \end{bmatrix} \in \mathbb{R}^{2 \times d_{\text{model}}}.$$

This 2-token sequence is processed by a stack of $L$ transformer blocks $\{\mathcal{F}^{(\ell)}\}_{\ell=1}^{L}$, each consisting of multi-head self-attention followed by a feedforward MLP, both wrapped in residual connections and layer normalization. Given an input $X \in \mathbb{R}^{2 \times d_{\text{model}}}$,

each block performs:

$$\tilde{X} = \text{LN}_1(X),$$
$$H = \text{MHSA}(\tilde{X}),$$
$$X' = X + H,$$
$$\hat{X} = \text{LN}_2(X'),$$
$$G = \text{MLP}(\hat{X}),$$
$$X^{\text{out}} = X' + G.$$

The multi-head self-attention over the two tokens is defined as follows. For each head $i = 1, \ldots, H$,

$$Q_i = XW_i^Q, \qquad K_i = XW_i^K, \qquad V_i = XW_i^V,$$

with $W_i^Q, W_i^K, W_i^V \in \mathbb{R}^{d_{\text{model}} \times d_k}$, and

$$\text{Attn}_i(X) = \text{softmax}\left(\frac{Q_i K_i^\top}{\sqrt{d_k}}\right) V_i \in \mathbb{R}^{2 \times d_v}.$$

The per-head results are concatenated and projected:

$$\text{MHSA}(X) = \big[\text{Attn}_1(X); \ldots; \text{Attn}_H(X)\big] W^O,$$

with $W^O \in \mathbb{R}^{(Hd_v) \times d_{\text{model}}}$. Because the sequence length is fixed at exactly two tokens, the transformer effectively learns how to mix and combine information between the memory token and the observation token, creating a learned recurrent update.

After passing through all $L$ blocks, we obtain:

$$X_t^{(L)} = \begin{bmatrix} h_t^0 \\ h_t^1 \end{bmatrix} \in \mathbb{R}^{2 \times d_{\text{model}}}.$$

We take the second token as both the output representation and the next-step memory state:

$$\tilde{h}_t = h_t^1 \in \mathbb{R}^{d_{\text{model}}}.$$

At episode boundaries, memory is reset using the done flag, following the implementation of LSTM by (Huang et al., 2022), $d_t \in \{0, 1\}$:

$$\tilde{h}_t \leftarrow (1 - d_t)\, \tilde{h}_t.$$

On the output side, the same three heads as in the PPO-LSTM architecture operate on the transformer hidden state $h_t$. Thus, relative to the PPO-LSTM agent, the only architectural change lies in the recurrent core: instead of an LSTM's gated updates, the PPO-RMT performs a learned recurrence via self-attention over a fixed pair of tokens representing the previous memory and the current observation.

### E.2. Results

**Table S13. Performance and language statistics for PPO-RMT.** Values are reported as mean $\pm$ standard deviation.

| Model | $N_{\text{pop}}$ | Self-SR | Cross-SR | IC | LS |
|-------|------|---------|----------|-----|-----|
| XP    | 2 | $0.400 \pm 0.121$ | $0.951 \pm 0.000$ | $0.420 \pm 0.127$ | $0.354 \pm 0.036$ |
| XP+SP | 2 | $0.953 \pm 0.003$ | $0.951 \pm 0.005$ | $1.002 \pm 0.002$ | $0.636 \pm 0.020$ |
| XP    | 3 | $0.939 \pm 0.009$ | $0.942 \pm 0.007$ | $0.997 \pm 0.002$ | $0.631 \pm 0.024$ |

We conduct experiments in *ScoreG.* and compare the results with PPO-LSTM. The first part is a qualitative comparison, showing that self-play and population training can lead to a shared, interchangeable language. The second part is comparing cultural transmission performance between PPO-LSTM and PPO-RMT.

**PPO-RMT agents show that self-play and larger populations improve language interchangeability.** Across conditions, XP agents display low *Self-SR* ($0.40 \pm 0.12$) despite strong *Cross-SR*, indicating poor interchangeability and low *LS*. Adding self-play (XP+SP) substantially improves performance, yielding high *Self-SR* ($0.953 \pm 0.003$), stable *Cross-SR*, and increased *LS*. Increasing the population size without self-play (XP with $N_{pop} = 3$) similarly enhances language consistency and interchangeability, achieving high *Self-SR* ($0.939 \pm 0.009$) and strong *LS*. Overall, both self-play and larger populations support more consistent and interchangeable communication in PPO-RMT agents.

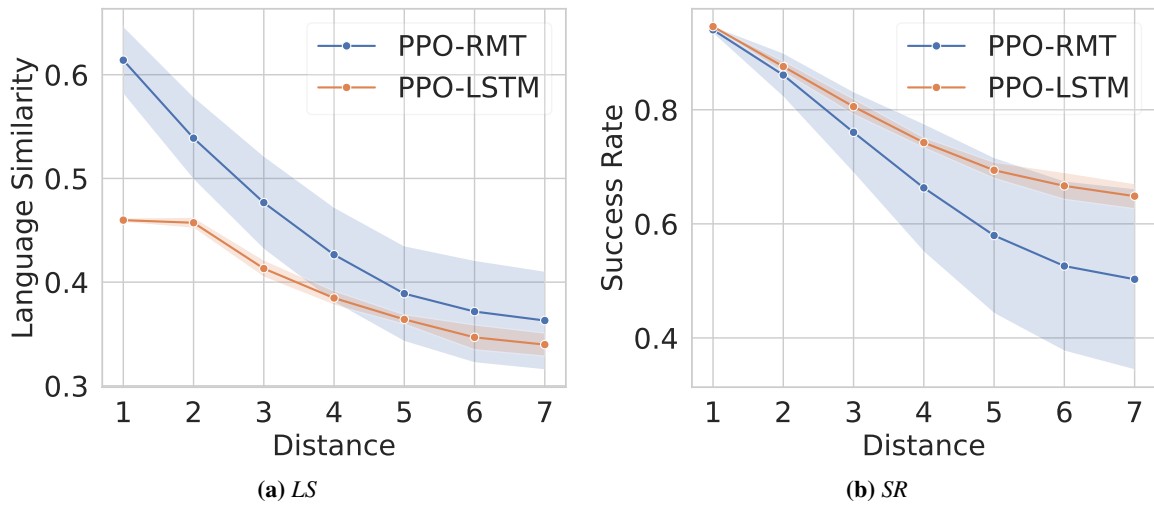

(a) *LS*                    (b) *SR*

**Figure S13. PPO-RMT vs. PPO-LSTM trained with XP+SP under a Ring Social Network.** (a) Language Similarity (*LS*) across social distances. (b) Success Rate (*SR*) across social distances. Shaded areas denote standard deviations across 3 training seeds.

**PPO-RMT exhibits larger variance in LS and SR than PPO-LSTM.** As shown in Figure S13, both PPO-LSTM and PPO-RMT trained with XP+SP display gradually decreasing *LS* and *SR* as social distance increases, indicating that self-play facilitates cultural transmission to non-partner (unseen) agents. However, PPO-RMT shows noticeably higher variance in both *LS* and *SR*, suggesting that cultural transmission under PPO-RMT is less stable and more sensitive to training seeds compared to PPO-LSTM.

