# OpenReview forum: "From Grunts to Lexicons: Emergent Language from Cooperative Foraging"
_ICML.cc/2026/Conference — Submitted to ICML 2026_

### Official Review · Reviewer_sXx3 · 2026-03-10

**Soundness:** 2
**Presentation:** 2
**Significance:** 2
**Originality:** 3
**Overall Recommendation:** 3
**Confidence:** 4

**Summary:**

This paper presents Foraging Games, a partially observable multi-agent environment for studying the emergence of communication under cooperation. The study shows agents develop languages with linguistic properties such as interchangeability, cultural transmission, displacement, and aspects of compositionality. It further examines how training regimes, population size, and social-network structure shape these emergent protocols and decodes spatial and temporal information from messages.

**Compliance With Llm Reviewing Policy:**

Affirmed.

**Final Justification:**

My concerns remain, so I keep my score.

**Key Questions For Authors:**

* Could the authors clarify why comparisons with large language model-driven agents were omitted, and whether such architectures might offer advantages for the studied tasks?
* Would it be possible to reorganize the results around the defined research questions (Q1–Q5) to improve clarity and readability?
* Can the authors better highlight the paper’s contributions and discuss the real-world significance of their findings from the agent society experiments?

**Limitations:**

yes

**Strengths And Weaknesses:**

### Strengths
* Provides a flexible testbed for studying embodied coordination and how social dynamics shape shared protocols.
* Shows how self-play and population size/connectivity affect interchangeability and cultural transmission, offering insights for MARL and EC research.
* The limitations of the current model and experimental design are well discussed in the Limitations section.

### Weaknesses
* Related work largely omits comparisons with agents driven by large language models and does not justify why such an architecture is unnecessary.
* The "a drop of 6% in Self-SR" in Table 1 is puzzling.
* Lacking a full comparison to alternative training methods to contextualize the benefits and costs of decentralization.
* The results section could be more structured. Since the study defines distinct research questions (Q1–Q5), organizing the findings around each question would improve clarity and make the narrative easier to follow.
* While the authors provide extensive descriptions of the agent society experiments, they do not clearly articulate the paper’s contributions, particularly regarding their implications for real-world study.

---

> ### Author Rebuttal · Authors · 2026-03-31
>
> We thank ```Reviewer sXx3``` for the thoughtful review and helpful suggestions on presentation and positioning. **Our work is intended as a scientific study of emergent communication in embodied multi-agent settings, rather than as a benchmark against MARL or LLM-based systems.** We will clarify this framing more explicitly in the revision and address each point below.
> ***
> ## sXx3-W1. Why we do not compare against LLM-driven agents
>
> Our goal is to study how communication protocols **emerge from scratch** under cooperative and ecological pressures. **Comparing against pretrained LLM-based agents would change the scientific question**, since such models already encode modern human language. LLM agents may be useful for improving task performance in embodied coordination, but they are much less suitable for studying how language-like structure itself arises.
>
> **Our framing follows the well-established Emergent Communication literature**, where agents develop protocols without access to pre-existing language (Lazaridou et al., 2017; Boldt et al., 2024; Gualdoni et al., 2024). We will clarify this distinction more explicitly in the Related Work and Discussion.
> ***
> ## sXx3-W2. Clarification of the “6% Self-SR” in Table 1
> **This result is one of the paper’s key findings about interchangeability**.  In the XP (N_pop=2) setting, agents are never exposed to their own messages during training. They develop asymmetric protocols: Agent A produces L1 for Agent B to hear but expects to receive L2 in return, and vice versa. Because agents do not learn to understand their own messages, they fail when paired with copies of themselves at test time (Line 308). This is analogous to an OOD generalization in ML.
> We will revise the text around Table 1 to make this interpretation explicit and avoid the impression that it is an anomaly or flaw.
> ***
> ## sXx3-W3. Why we use decentralized training
> Our use of decentralized, non-parameter-sharing training is a deliberate modeling choice tied to the scientific goal of the paper. **Our goal is to simulate ecologically valid conditions under which human-like communication can emerge among heterogeneous agents**, where individuals do not share any parts of their brains.
>
> **We adopt fully decentralized training to reflect the autonomy of biological agents**, which lack shared parameters and access to others’ internal states.
> In contrast, state-of-the-art methods in MARL **typically rely on parameter sharing** (Amato, 2024), giving agents a common brain and a shared language from the outset. This makes them **unsuitable for studying language heterogeneity**, cultural transmission, and interchangeability. We will clarify these motivations in the final version.
>
> ***
> ## sXx3-W4. Structuring the results around Q1–Q5
> We agree that restructuring the results around Q1–Q5 will improve readability. The five highlighted findings in bolded text in the result section (Section 5) are designed to map to these questions. In the final version, we will explicitly label each finding to directly correspond to its respective research question.
> ***
> ## sXx3-W5. Contributions and broader significance of the agent-society experiments
> We thank the reviewer for this suggestion. We will revise the Discussion and Future Work sections to state the paper’s contributions and broader implications more clearly. In particular, we will emphasize that:
> ### Scientific contribution
> Our main contribution is a framework for studying how communication structure emerges in embodied cooperative populations under ecological and social pressures. It also contributes to computational studies of language evolution. The findings provide insight into how heterogeneous agents develop, align, and transmit communication protocols through interaction.
>
> ### Potential practical domains
> These ideas could be extended to the applications of multi-robot coordination, human–AI collaboration, and other settings where agents must develop efficient task-specific communication, like autonomous driving and cooperative perception (Cui et al., 2022).
>
> ***
> ## sXx3-Q1. Clarify why comparisons with LLM agents were omitted.
> We address this point in ```sXx3-W1```.
> ***
> ## sXx3-Q2. Results reorganized around Q1–Q5?
> We address this point in ```sXx3-W4```.
> ***
> ## sXx3-Q3. Can the contributions and broader significance be highlighted more clearly?
> We address this point in ```sXx3-W5```.
> ***
> ## Reference
> - Lazaridou et al. (2017), Multi-Agent Cooperation and the Emergence of (Natural) Language, ICLR
> - Boldt et al. (2024), A Review of the Applications of Deep Learning-Based Emergent Communication, TMLR
> - Gualdoni et al. (2024), Bridging semantics and pragmatics in information-theoretic emergent communication, NeurIPS
> - Amato (2024), An Introduction to Centralized Training for Decentralized Execution in Cooperative Multi-Agent Reinforcement Learning
> - Cui et al. (2022), End-to-End Driving with Cooperative Perception for Networked Vehicles, CVPR

---

> > ### Author Rebuttal · Reviewer_sXx3 · 2026-04-02
> >
> > Thanks for the rebuttal. My two main remaining concerns are the mismatch between the paper’s narrow setting and its broader claims, and the limited justification of its practical significance:
> >
> > 1）While I understand the authors’ intention to study communication emerging from scratch, this does not fully resolve my concern. The paper still uses broad language-oriented framing, including human-like communication, ecological validity, and real-world relevance, while the actual system remains a minimal task-specific protocol. If the goal is to study protocol emergence in a controlled setting, the contribution should be framed more narrowly. If the goal is to draw broader conclusions about language-like structure or practical communication, then the paper should better justify why language-capable agents are not even a relevant reference point.
> >
> > 2）A related concern is practical significance. Since the emergent protocol here is highly simplified and far from human language, it remains unclear what concrete insight this framework offers for applications such as multi-robot coordination, human-AI collaboration, or real-world multi-agent communication. This is especially important because LLM-driven agents are already beginning to show practical advantages in these settings, with communication grounded in human language often being more interpretable, easier to integrate, and potentially more useful in deployment. At present, I find the work more convincing as a toy model of emergent communication than as a framework with clear downstream relevance.

---

> > > ### Author Response · Authors · 2026-04-02
> > >
> > > Thank you again for the thoughtful follow-up. We agree that the framing should match the scope of the evidence, and we are happy to tone down broader wording, such as **ecological validity**, in the revision. In the current submission, **we already note in the Limitations section that the emergent language in our setting remains far from natural language (```Line 407 right, Line 423 right```).** We also **did not intend to claim human-like communication**. As suggested by the reviewer, we will make it more explicit by mentioning in **Introduction** in the final version. We would therefore like to clarify the remaining points in detail regarding the paper’s framing and broader significance.
> > >
> > > ***
> > > ## Framing and terminology.
> > > To address this directly:
> > >
> > > 1. We will remove Emergent Language from the title and revise it to something along the lines of
> > > **Emergent Communication with Linguistic Properties in Cooperative Foraging Games.**
> > > 2. We will revise the terminology throughout the paper by changing Emergent Language to Emergent Communication in the main framing.
> > >
> > > We believe this better reflects the actual scope of the evidence while preserving the paper’s central contribution.
> > > ***
> > > ## Justifying why we do not compare with LLMs.
> > > The main reason for not comparing to language agents is that the emergent communication (in most cases) is narrow (task-specific) and far from human language. We note that **comparison to LLM-based agents is not a typical requirement in this line of work, please see for example [1-6]**. Much of the literature studies either (i) how task-specific communication emerges from scratch, or (ii) how learned communication improves coordination under practical constraints. In both cases, the central question is usually not how a pretrained full-language system would perform on the same task. **We really appreciate the recommendation for adjustment in framing, and we will clarify this reason in the final version.**
> > >
> > > ***
> > > ## Clarifying language-like / linguistic properties
> > > By linguistic / language-like properties, we do not mean full human-like communication. Rather, we refer to functional design features discussed in the literature, such as those associated with **Hockett’s design features** [8]. Importantly, the presence of some such properties does not imply equivalence to human language. For example, **animal communication systems can exhibit certain language-like properties [7] while remaining far from human language** in richness, flexibility, and semantics. For this reason, we do not view comparison to LLMs as the most relevant reference point for the present paper.
> > >  We will put this justification in the final version, as suggested by the reviewer. We are also happy to change the term **language-like properties** to **linguistic properties**, which is clearer for the broader audience.
> > > ***
> > > ## Practical significance.
> > > We fully agree that the paper is stronger as a controlled scientific framework than as an immediate application paper, and **we will revise the discussion to make that clearer.** We therefore see the practical value as indirect and foundational rather than deployment-ready. This is also consistent with the paper’s existing limitations: **it already states that the results should be interpreted as functional analogues** (```Line 406, right column```), that the emergent protocol is simple and not directly comparable to natural language, and that the work is **purely theoretical with no immediate deployment claim** (```Line 445, left column```).
> > >
> > >
> > > ***
> > > ### Reference
> > >
> > > - [1] Multi-Agent Reinforcement Learning with Communication-Constrained Priors (NeurIPS'25)
> > > - [2]  Learning Multi-Agent Communication with Contrastive Learning (ICLR'24)
> > > - [3] Learning Multi-Agent Communication from Graph Modeling Perspective (ICLR'24)
> > > - [4]  Speaking Your Language: Spatial Relationships in Interpretable Emergent Communication (NeurIPS'24)
> > > - [5] CtD: Composition through Decomposition in Emergent Communication (ICLR'25)
> > > - [6] TACTIC: Task-Agnostic Contrastive pre-Training for Inter-Agent Communication (AAMAS'25)
> > > - [7]  The syntax–semantics interface in animal vocal communication
> > > - [8] The origin of speech

---

### Official Review · Reviewer_4FCv · 2026-03-11

**Soundness:** 3
**Presentation:** 2
**Significance:** 3
**Originality:** 3
**Overall Recommendation:** 5
**Confidence:** 3

**Summary:**

This paper investigates the emerging communication between agents in two different, but related gaming settings. The experiments are reinforcement learning environments in which agents do not only see and interact with the environment, but also with one another, through messages from a pre-defined vocabulary. Their interaction networks are varied in topology. Features of natural language are then measured. The authors investigate the emergence of natural language features as a function of population size, communication network topology and temporal constraints.

**Compliance With Llm Reviewing Policy:**

Affirmed.

**Final Justification:**

When writing my original review, I did indeed have the experiments in mind that the authors performed to write the rebuttal. These early results seem to potentially weaken the results of the paper. The method would become more sound, but the results less significant. I am not against leaving this for future work, as a lot of computational effort needs to be done for a full suite of experiments, but the soundness-significance balance leaves me at the same judgement for the paper overall. The discussion on terminology is on the one hand understandable, as sticking to literature conventions makes it easier to read for intended audiences, but the problems that arise in the interpretive leaps caused by the terminology is a very serious issue for the whole field. The title and discussion should be considerably weakened in their terminology for reasons of honesty.

**Key Questions For Authors:**

1. How can you distinguish between emergent communication (which I think you convincingly show properties of) and emergent language (which you seem to be more posed and implicated)?
2. The entropy measures in the different experiments seem to all converge quickly, but why would that be sufficient for convergence of all properties for all network topologies?
3. How does this numerical setup scale to larger environments, and more complex tasks for the agents?

**Limitations:**

One of the main limitations of this paper as I see it is the overselling of results. The results are intersting in themselves and teach us plenty about emergent communication, but the extrapolation of those results into emergent language is too large. Even though the experiments are well set up and sound, their compuational domain is very limited, which rises questions about scaling and generalizability.

**Strengths And Weaknesses:**

**Soundness**.
The setup of the experiments is sound and well described. The parameter space investigated is large and encompasses much of the interesting variation. One could argue that the grid, and especially the observable part of it is small, which may limit generalizability of the results.

I applaud the convergence experiments shown in the supplemental material, but I do have concerns about convergence of other properties. Particularly, the results in Fig 3, where the language parameters are shown as a function of distance between agents in the social network, could be sensitive to limited numbers of cycles (a shared language takes longer to settle over many different steps of communication, even if the shown entropies have largely converged).

**Presentation**.
The paper is written well. The experiments follow nicely from the introductory section and are clearly described, followed by a well-structured discussion. The figures would benefit from larger labels.

One major issue I have is with the title, and the interpretation of the results as "emergent language". The experiments are convincing, but show only that there is emerging meaningful communication. Animals, and even plants, also communicate meaningfully, but they don't exhibit a language (in most cases, anyway). The same is true for the agents presented in the experiment. They have been given a set of signals they can send to one another. Calling that set a vocabulary doesn't make the signals get a semantic meaning. I would have preferred it strongly if the authors would have stuck to calling it "emergent communication", as they sometimes do in the paper, throughout, and especially in the title, which in its current form oversells the results.

The same holds, to an extent, for terminology like "cultural transmission", which I think is a huge interpretative leap from the phenomenon simulated in the paper.

**Significance**.
The work is a significant contribution in the growing field of numerical studies into emerging communication and language development. The introduced Foraging Games are a significant contribution to the field, and the setup of experiments fosters a numerical experiment that has a lot of potential for the development of a solid framework for the numerical investigation of emergent communication in a wide variety of settings.

For sound conclusions on meaningful communication, and hence language development, though, there needs to be more. Larger populations of agents, which independently develop very similar communication patterns would make the results of this paper more significant, although I recognize this as a first step in a promising direction. It is as yet not clear to the reader how the numerical experiments would scale to more relevant grid and population sizes.

**Originality**.
The paper is well grounded in related literature and adds a significant, and original contribution. The Foraging Games underline the social interaction, and the addition of self-play seems crucial for reproducible and understandable emergent communication.

---

> ### Author Rebuttal · Authors · 2026-03-31
>
> We thank ```Reviewer 4FCv``` for the positive and thoughtful review. We really appreciate the helpful comments on convergence, terminology, and scaling, and we address each of these points below.
> ***
> ## 4FCv-W1. Convergence of properties beyond return and entropy
> Stable episode return (Figure S1(a)) is a standard indicator that a population has reached a near-optimal cooperative strategy. However, the reviewer raised a fair point that communication protocols may remain in flux even after the return converge. To ensure the stability of the emergent communication, **we monitor the approximate KL divergence** between successive policy updates throughout training. As training progresses, this KL divergence approaches zero, confirming that the communication policy is no longer changing. This ensures that the patterns observed in Figure 3 represent settled states, rather than snapshots of evolving policy. We will clarify these criteria and include the KL-div plot in the final version.
> ***
> ## 4FCv-W2. Terminology: emergent communication vs emergent language
>
> We agree that communication and language can be interpreted at different levels of strength. Our use of the term **emergent language** (EL) follows standard usage in the field (Lazaridou et al., 2017; van der Wal et al., 2020; Boldt and Mortensen, 2024, 2025), where it is often used interchangeably with **emergent communication** (EC) (Gualdoni et al., 2024) to describe communication systems that arise from interaction and exhibit language-like properties.  That said, the reviewer is right that EL is not a full-fledged natural language (and we do not claim that it is), although these share some similar properties.
>
> Similarly, our use of **Cultural Transmission** is meant in the limited multi-agent sense of conventions or behavioural repertoires being transmitted across agents and social structure (Bhoopchand et al., 2023; Cook et al., 2024), not in the full richness of human cultural processes. We will clarify both points more explicitly in the final version.
> ***
> ## 4FCv-W3. Scaling to larger populations and environments?
> We agree that the emergence of similar, interchangeable communication patterns in large populations is a hallmark of robust language development. To address this, **we conducted a new experiment increasing the population size to 100 agents**, trained using pure Cross-Play (XP), where agents are never paired with themselves during training.
>
> In Table III, agents in a population of 100 achieve a Self Success Rate (Self-SR) of 96.8%, which is significantly higher than the 2-agent baseline. While Language Similarity (LS) is lower than in the 3-agent case, likely due to the increased difficulty of aligning a large decentralized group, it remains well above the 2-agent baseline.
>
> ### Table III: 100-Agent Convergence
> | Num Agents | Cross-SR | Self-SR | LS |
> |---|---|---|---|
> | 2 | 98.7 | 6.5 | 0.215 |
> | 3 | 97.7 | 94.4 | 0.527 |
> | 100 | 96.7 | 96.8 | 0.364 |
>
> Scaling to **larger grids and more complex tasks** is also feasible, but in practice it requires more careful optimization. Training on larger environments is substantially more difficult from scratch due to the large search space, and **curriculum learning** from smaller grids to larger ones may be a useful strategy. Moreover, one **could relax some biologically plausible constraints** in our setup to let the agents **exchange gradients** during learning, speeding up the optimization.
>
> We will clarify this point in the revision and expand the discussion of scaling limitations and the future work.
> ***
> ## 4FCv-Q1. Distinguish between emergent communication and emergent language
>
> We view emergent communication as the broader phenomenon of agents developing meaningful signaling systems through interaction. We use **emergent language** in the standard sense used in the field: emergent communication systems that exhibit language-like properties such as discreteness, compositionality, interchangeability, and displacement. We do not claim that these protocols are equivalent to human natural language, and we will clarify this distinction more explicitly in the revision.
> ***
> ## 4FCv-Q2. Why is entropy enough?
> We address this in 4FCv-W1.
> ***
> ## 4FCv-Q3. How does this setup scale to larger environments and more complex tasks?
> We address this in  4FCv-W3.
> ***
> ## Reference
> - Lazaridou et al., Multi-Agent Cooperation and the Emergence of (Natural) Language, ICLR’2017
> - van der Wal et al., The Grammar of Emergent Languages, EMNLP’2020
> - Boldt and Mortensen, Searching for the Most Human-like Emergent Language, EMNLP’2025
> - Boldt and Mortensen, A Review of the Applications of Deep Learning-Based Emergent Communication, TMLR2024
> - Gualdoni et al. (2024), Bridging semantics and pragmatics in information-theoretic emergent communication, NeurIPS
> - Bhoopchand et al., Learning few-shot imitation as cultural transmission, Nature Com. 2023
> - Cook et al., Cultural Accumulation in Reinforcement Learning, NeurIPS’2024

---

> > ### Author Rebuttal · Reviewer_4FCv · 2026-04-03
> >
> > Convergence: I explained my concerns about convergence around Fig S1(a), and including the material explained by the authors in the rebuttal would take that  concern away. Convergence in such numerical experiments is crucial, so a broader investigation of convergence against all numerical resolution elements is still very much worth doing.
> >
> > Terminology: I am aware that authors choose to stay close to terminology used in literature around the topic. Using the terminology as is in describing properties of agents is OK. Where my problem mostly lies is that tis terminology gets extrapolated into discussion, interpretation and even the title of the paper. The interpretative leaps taken here are easy to understand given the terminology (hence my issue with the terminology), but not at all justified by what is tested in the experiments. The authors rightfully claim they don't simulate the emergence of a full-fledged natural language, but could any reader guess that from the title alone?
> >
> > For the use of the term "language", the authors claim that the communication system exhibits discreteness (I agree), compositionality (I don't think that's shown), interchangability (partially agree), and displacement (disagree).
> >
> > Scaling: The extra experiment shown in the rebuttal is relevant, and it would be good to include such details in the paper. The lower language similarity is then something to carefully investigate: is it due to the difficulty in aligning a large decentralized group (again, this hints at convergence issues), or is the emergence of the langauge not as strong as suggested in a very-few agent setting might suggest? The added experiment here is a step in the right direction, but whether this stengthens or in fact wekeans the results of the paper is far from clear at this point.

---

> > > ### Author Response · Authors · 2026-04-04
> > >
> > > Thank you again for the thoughtful follow-up and for highlighting these interpretation issues so clearly.
> > >
> > > ### Terminology
> > > We take the reviewer’s point that the overclaim risk lies in the title and the surrounding interpretation.
> > > To address this,
> > > - (1) We will **remove Emergent Language** from the title and revise it to something along the lines of **Emergent Communication with Linguistic Properties in Cooperative Foraging Games.**  We are also open to the reviewer’s recommendation regarding alternative titles.
> > > - (2) We will revise the terminology throughout the paper by changing **Emergent Language** to **Emergent Communication** in the main framing.
> > >
> > > We believe this better reflects the actual scope of the evidence, while preserving the paper’s central contribution.
> > >
> > >
> > > ### Scaling Population
> > > We agree that the lower LS at population size 100 should be interpreted carefully. In a fully decentralized setting, some increase in heterogeneity is expected as population size grows, since aligning message forms across many agents is harder than in small populations. Importantly, Cross-SR and Self-SR remain high, indicating that agents still understand both one another and themselves despite reduced surface-level similarity. We therefore interpret this as evidence of more heterogeneous yet still functionally compatible communication at a larger scale, rather than simply weaker emergence. This interpretation is also **broadly consistent with prior work** in autoencoder settings, where **increasing population size can reduce language similarity without eliminating communicative success** (see ```Figure 4``` in Kim and Oh, 2021). **We will include this discussion in the final version.**
> > >
> > > ### Convergence
> > > Thank you for the helpful point. We agree that return and entropy alone are not sufficient. The additional KL-based convergence analysis directly resolves this concern by showing that the communication policy itself has stabilized, and **we will include the full figures in the final version.**
> > >
> > >
> > > ### Reference
> > > Kim and Oh, Emergent Communication under Varying Group Sizes and Connectivities, NeurIPS'21

---

### Official Review · Reviewer_7pA3 · 2026-03-13

**Soundness:** 3
**Presentation:** 3
**Significance:** 3
**Originality:** 2
**Overall Recommendation:** 4
**Confidence:** 3

**Summary:**

This paper investigates the emergence of communication in foraging games under a population setting. The author has introduced a foraging game (FG) framework for studying emergent communication. They propose a hybrid cross-play and self-play training regime that improves the cultural transmission across social networks. Experimental studies show that agents are able to engage in emergent communication that handles tasks with temporal or spatial displacement.

**Compliance With Llm Reviewing Policy:**

Affirmed.

**Final Justification:**

The authors have addressed my concerns and questions. I believe this work would benefit from including an analysis of the Vis-Com setting, as the authors have emphasised embodied communication. I am keeping my current score.

**Key Questions For Authors:**

1.	Did you consider the visible partner with communication (Vis-Com)? What is the SR, and how does it affect the language quality?
2.	For the XP+SP setting, what is the probability of selecting the agent itself, 50% or proportional to the population size?

**Limitations:**

yes

**Strengths And Weaknesses:**

Strengths
The paper is well-written and easy to follow. The authors have conducted an in-depth analysis of FG with emergent communication under various settings, including the language similarity across distances between agents in a network and the topographic similarity across population size.
The proposed cross-play and self-play approach has demonstrated improvement in the language similarity and success rate.

Weaknesses
1.	The idea of cooperative foraging with communication is not entirely novel; see, for example, [1]. The authors should compare with these works and discuss why their setting is preferable (like higher language similarity, more effective in a large population, fewer restrictions, etc.).
2.	The invisible partner setting of the game lacks justification. It is natural to assume that the agent can observe their partner when they are close enough. As the authors have emphasised embodied communication, it is also important to consider visual communication.
3.	The level of difficulty of the game (5x5 grid) is insufficient to demonstrate the improvement between inv-Com and Vis-NoCom. When the game is simple enough, cooperation can be achieved with non-verbal communication, and verbal communication may be poorly developed. This may be the reason for considering the invisible partner setting. It is suggested that the authors can introduce a more difficult game, so that cooperation must be achieved by both non-verbal and verbal communications.

[1] Azmani et. al. Cooperative Foraging Behaviour Through Multi-Agent Reinforcement Learning with Graph-Based Communication

---

> ### Author Rebuttal · Authors · 2026-03-31
>
> We thank ```Reviewer 7pA3``` for the thoughtful and constructive review. The suggestions on prior work, partner visibility, and task difficulty are very helpful, and we have addressed each of them with additional clarification and new experiments.
>
> ***
> ## 7pA3-W1. Novelty relative to prior cooperative foraging work (Azmani et al.)
> While both papers study cooperative foraging, they address different scientific questions:
> ### Our main novelty.
> Our novelty is not cooperative foraging per se, but using it as a testbed for emergent language in decentralized embodied populations. This framing lets us study linguistic properties such as **interchangeability, compositionality, cultural transmission, and temporal/spatial displacement**, which are not the focus of prior foraging-based MARL work.
> ### Different objective
> Azmani et al. focus primarily on coordination efficiency in a MARL setting, whereas our work uses foraging as an ecologically grounded testbed for emergent communication (Lazaridou et al.,2017; Mordatch and Abbeel, 2018). Our goal is to study **emergent linguistic properties** under social and ecological pressures that approximate some of the conditions under which early human language may have evolved.
> ### Communication is necessary in our setting
> We intentionally impose **knowledge asymmetry**, so each agent observes only part of the task-relevant information, and communication is required for success.
> ### Fully decentralized communication
> Unlike approaches based on shared hidden states, graph communication, or parameter sharing, our agents are fully decentralized and must develop discrete communication protocols from scratch, **without shared parameters or gradients**.
> We will add a discussion of Azmani et al. to the Related Work and clarify this distinction in the revision.
>
> ***
> ## 7pA3-W2. Why we use the invisible-partner setting
> The invisible-partner setting is a design choice to **isolate verbal communication from non-verbal visual communication**. If agents can observe one another, they may coordinate through body movements, which reduces the pressure to communicate through discrete messages.
> To verify this point, we conducted an additional experiment comparing our default **Inv-Com** (invisible partner) setting with a **Vis-Com** (visible partner) setting. Following the reviewer’s suggestion, we evaluated the structure and informativeness of the emergent protocol using TopSim and linear decoding of message embeddings into ground-truth item attributes (score, vertical position (V-Pos), and horizontal position (H-Pos)).
> We find that when partners are visible, **TopSim decreases from 0.47 to 0.33**, and **decoding accuracy also drops slightly**. This supports our design choice: visual access provides an alternative communication channel, thereby weakening the pressure for agents to encode task-relevant information in explicit messages. We will include these results in the final version.
>
> ### Table I: Comparison of Invisible vs. Visible Partner Settings
> | Model | Success (SR) | Acc: Score | Acc: V-Pos | Acc: H-Pos | TopSim | RepCom |
> | :--- | :--- | :--- | :--- | :--- | :--- | :--- |
> | **Inv-Com** | 0.968 ± 0.006 | 0.340 ± 0.02 | 0.746 ± 0.03 | 0.275 ± 0.007 | 0.47 ± 0.04 | 1.65 ± 0.04 |
> | **Vis-Com** | 0.970 ± 0.002 | 0.3102 ± 0.006 | 0.640 ± 0.01 | 0.246 ± 0.003 | 0.33 ± 0.03 | 1.60 ± 0.02 |
>
> ***
> ## 7pA3-W3. Harder environments and the role of visual signaling
>
> To address the concern that the original 5x5 setting may be too simple, we conducted additional experiments in a harder environment by (1) increasing the spatial search space from 25 to 49 cells (using 7x7), (2) adding obstacles, (3) expanding to a diverse population of $N_{pop}​ = 15$, and (4) restricting verbal communication to 6 steps, fewer than the total steps typically required to solve the task.
>
> In this more challenging environment (Table II), Inv-Com achieves only 59.3% success rate, whereas Vis-Com maintains 96.5%. This result shows that as environmental and social complexity increase, visual access to a partner becomes substantially more useful for coordination. At the same time, **our invisible-partner setting remains a deliberate control for isolating explicit verbal communication from non-verbal signaling**. We will include these in the final version.
>
> Table II:
> | Policy | SR (%) | Avg. Length
> | :--- | :---: | :---: |
> | Vis-Com | 96.5 | 8.3 |
> | Inv-Com | 59.3 | 11.2 |
> ***
> ## 7pA3-Q1. Did you consider the Vis-Com condition?
> We address this in 7pA3-W2.
> ***
> ## 7pA3-Q2. Probability of selecting agents in XP+SP
> We select agent pairs $(a_1, a_2)$ uniformly from the set of all possible combinations, including self-pairs. For example, when $N_{pop} = 3$, we choose from the nine possible pairs {[1,1], [1,2], [1,3], ..., [3,3]\} with equal probability. If the population size is defined as N, the probability of self-play is 1/N. We will clarify this sampling procedure in the final version.

---

> > ### Author Rebuttal · Reviewer_7pA3 · 2026-04-04
> >
> > Thank you for the authors’ reply.
> >
> > The authors have addressed my concerns and questions. I believe this work would benefit from including an analysis of the Vis-Com setting, as the authors have emphasised embodied communication. I am keeping my current score.

---

### Decision · Program_Chairs · 2026-04-30

**Decision:**

Reject

**Comment:**

The paper presents an analysis of emergent language in the context of cooperative foraging games. The setting is grid worlds with hidden rewards and temporal dependencies. The results are interpreted as demonstrating the hallmarks of language: arbitrariness, interchangeability, displacement,cultural transmission, and compositionality. These are analyzed against a collection of parametric variations.

Reviewers commended the paper for its readability, the detailed analysis of parametric variations, and in the flexibility of the paradigm.

These strengths were, however, outweighed by the weaknesses. First, the gridworld paradigm, while flexible and amenable to parametric variation, lacks ecological validity. Second, there were concerns about overinterpretation of the results that were not satisfyingly addressed. The conclusions are extended to langauge, but it is unclear what this environment can say about language in general. The authors indicate the willingness to weaken this claim to communication, but the same concerns remain. Similarly, claims are related to embodiment, but these are also overstated. It would be important to extend this work to richer contexts to ensure that the results are ecologically valid.